# FSSM: FREQUENCY-SELECTIVE STATE-SPACE MODELS FOR SPECTRAL REPRESENTATION LEARNING

## ABSTRACT

We introduce the first state-space model (FSSM) with frequency selective spectral operators, parameterizing a family of stable, causal, band-selective kernels whose spectral weights are conditioned on the end task. This yields a representation that adapts its characteristics per task domain while retaining linear-time inference and memory. The key novelty is the trainable spectral front-end through which the model can adapt frequency weighting and inter-bin window size. We show the effectiveness of our learned spectral representations on two independent domains: radar object detection and speech keyword recognition, outperforming state of the art frequency based methods in both domains while maintaining competitive throughput and computational overhead. We further show the robustness of our approach under input perturbations, demonstrating the value of stabilized sequential operators in spectral representation learning.

## 1 INTRODUCTION

Short-term Fourier analysis or fixed filterbanks form the basis of various signal processing pipelines for time-series data with applications ranging from automotive radar to speech keyword spotting. While fast and interpretable, such front-ends are agnostic to downstream objectives: window sizes, frequency grids, and tapering are chosen *a priori* and remain static across instances and time. Recent learnable Fourier layers introduce trainable parameters but still operate as block transforms detached from sequential dynamics, limiting their ability to track nonstationary spectra or adapt frequency emphasis on the fly.

We propose the first *Fourier-initialized, frequency selective state-space module* (FSSM) that adaptively learns spectral representations while retaining linear-time, streamable inference (Figure 1. The module realizes a bank of damped rotating modes with backward-Euler stabilization and Fourier-like initialization, then conditions input contribution to emphasize task-relevant frequencies. Readouts are phase-aligned complex coefficients representing spectral features. For real inputs we exploit half-spectrum symmetry to avoid redundant computation, while for I-Q radar data we estimate channel spectra and combine them in the complex domain.

We demonstrate the efficacy of this design in two domains, namely, object detection from radar, and speech recognition. For high-definition radar, we show the flexibility of the proposed model by independently combining with convolution based [Rebut et al. (2022a)] and transformer based [Giroux et al. (2023)] decoder networks, and in both cases the model achieves state-of-the-art range-azimuth detection and a significant performance increase over traditional discrete Fourier transform (DFT) or learnable DFT encoders. For audio keyword detection results, we verify the robustness of our model over other spectral front-ends as a proof of concept.

The key contribution of the paper is as follows:

- The first analytical implementation of a Fourier-initialized multi-mode SSM that adaptively learns spectral features with linear-time, streamable inference.
- Domain-specific instantiations: dual-axis spectral learning for radar (range and Doppler) and a windowed, magnitude-only variant for audio integrated with downstream decoders.
- Achieving state-of-the-art radar detection and superior audio keyword recognition under synthetic noise compared to fixed FFT and learnable DFT baselines.

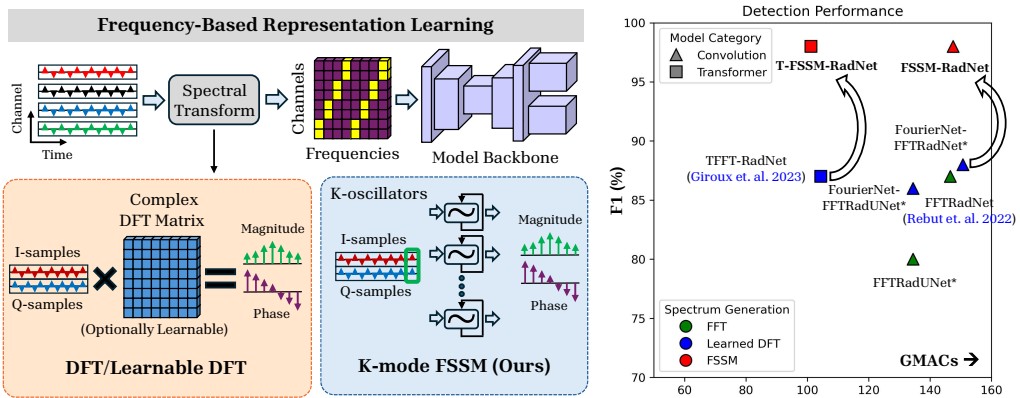

Figure 1: Traditional spectral representation vs our spectral representation learning approach with SSM (left). Our learning approach achieves state of the art performance on radar based object detection and freespace segmentation tasks (right).

## 2 BACKGROUND

**Frequency Modulated Continuous Waveform Radar Range and Doppler Relations.** In an FMCW radar, a radio wave is transmitted with time varying frequencies, repeated every $T_r$ seconds per radar frame (chirps):

$$s_{tx}(t) = e^{j2\pi(f_c t + \frac{st^2}{2})}$$

Where $f_c$: carrier frequency, $S = \frac{B}{T_c}$: chirp slope with bandwidth $B$ and chirp duration $T_c$. For any point reflecting target at range $R$ and radial velocity $v$:

$$\text{Round trip delay}, \tau = \frac{2R}{c}; \quad \text{Doppler frequency}, f_D = \frac{2vc}{f_c}$$

Here, c is the speed of electromagnetic wave propagation. The received signal is approximately,

$$s_{rx}(t) = \alpha s_{tx}(t - \tau)e^{j2\pi f_D t}$$

Where $\alpha$ is a complex attenuation term. After mixing and de-chirping, the received signal is I-Q sampled. I-Q sampling is done on 90° phase intervals, retaining the amplitude and phase information of the full spectrum of the incoming signal. The sampled signal (sampling period $T_s$) is approximately:

$$x[m, n] \approx \alpha e^{j2\pi(f_r nT_s + f_D mT_r)}$$

A DFT over sample and chirp axes respectively $(n, m)$ gives peaks at range frequency $f_r$ and doppler frequency $f_D$:

$$\text{Range}, R = \frac{cf_r}{2S}; \quad \text{velocity}, v = \frac{\lambda f_D}{2}$$

For multiple targets, the received signal is a sum of such 2D sinusoids, and the 2D DFT gives a range-Doppler map, where each bright point corresponds to a target at some $(R, v)$.

**Fourier Transform and Audio Spectrograms.** Speech is locally stationary over short windows, so the short-time Fourier transform (STFT) applies a windowed FFT:

$$X(t, \omega) = \sum_\tau x[\tau]\omega[\tau - t]e^{-j\omega\tau}$$

Sliding this window produces a spectrogram $|X(t, \omega)|^2$, a time-frequency image where columns capture local harmonic content and rows track its temporal evolution. Because spectrograms have strong 2D locality-formants, harmonics, onsets-CNNs and vision-inspired models operate effectively on them, while attention mechanisms capture longer-term temporal structure. The Fourier transform remains central because it reveals the frequency-varying structure of speech that is obscured in raw waveforms.

**Discrete Fourier Transforms and learnable-DFT front-ends.** A DFT assumes the signal fits exactly into one of its discrete frequencies $f_k = \frac{k}{N} f_s$. If the true tone is at a non-bin-centered frequency $f_0 \notin \{f_k\}$, then $x[n] = e^{j 2\pi f_0 n / f_s}$ does not yield a single DFT peak. Instead,

$$|X[k]| = |\sin(\pi(\frac{f_0}{f_s} - \frac{k}{N})N) / \sin(\pi(\frac{f_0}{f_s} - \frac{k}{N}))|$$

produces a broadened main-lobe and leakage into neighboring bins. This smearing affects radar sharpness(off-grid ranges/Dopplers spreading across bins, weakening peaks) and audio (blurred harmonics or poor time-frequency contrast) [Lyon (2009)]. Learnable spectral front-ends attempt to relax these assumptions-either by directly parameterizing the DFT/FFT (e.g., differentiable DFT layers, butterfly/structured unitary transforms, or Fourier-mixing modules) [Lee-Thorp et al. (2021)] or by learning filterbanks (e.g., SincNet with tunable sinc filters; LEAF with learnable Gabor-like filters) [Ravanelli & Bengio (2018); Zeghidour et al. (2021); Schlüter & Gutenbrunner (2022)]. Empirical observations indicate that learned filterbanks may converge to near-mel configurations or offer only marginal deviations, and they do not consistently outperform carefully designed fixed features under distribution shift [Schlüter & Gutenbrunner (2022)].

**state-space Models for sequence modeling.** Linear state-space models (SSMs) parameterize $x_{t+1} = Ax_t + Bu_t$, $y_t = Cx_t$, whose state retains memory of a very long horizon of input sequence. When discretized with stable parameterizations and implemented via parallel convolution or scan, SSM layers provide *linear-time*, streamable sequence processing with strong and controllable memory, offering an alternative to quadratic-time attention. Details on different initialization of SSM models have been discussed in Section A.3

**Positioning the present work.** We adopt the SSM perspective into the *spectral* front-end: our frequency-selective SSM learns band-selective kernels. This spectral representation learning preserves linear-time inference while adapting frequency emphasis and effective windowing per instance. On radar, it replaces fixed DFT stacks with learned, sequential spectral operators, improving clutter suppression and sensitivity to weak and far-range targets compared to Fourier-anchored baselines such as FFTRadNet and its transformer variants [Rebut et al. (2022b); Giroux et al. (2023)]. On speech commands, it replaces the STFT stream as a per-frame spectral emphasis before lightweight sequential decoders (e.g., Mamba Decoder), yielding robust performance under noise and temporal variation [Gu & Dao (2023)]. Together, these results indicate that input-selective spectral learning unifies the strengths of Fourier analysis and SSMs across modalities while maintaining modest computational and time budgets. (more in Section A.4).

## 3 Methodology

### 3.1 Frequency-Selective state-space Module

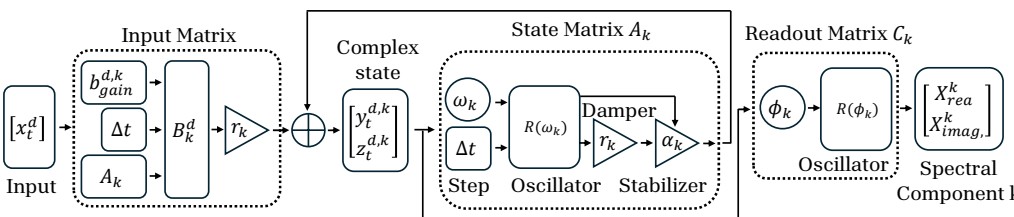

Figure 2: Our proposed FSSM methodology. Each mode updates the complex state vector using learnable rotation, magnitude damping and backward Euler stabilization with state-dependent input projections. Output is generated using another complex rotation matrix to get in-phase and quadrature components. Total K oscillators are used to find the K spectral components.

FSSM is a learnable DFT front-end: each mode is a damped complex sinusoid initialized to match a DFT bin, then trained to adapt its center frequency and bandwidth. We implement these modes as stable state-space recurrences for efficiency, but conceptually they are learnable DFT filters

---

**Algorithm 1** FSSM: Initialization and Streaming Update

---

**Input:** sequence $x[0{:}L-1]$, number of modes $K$.
 1: **Parameters (learnable):** $\theta, \rho, A^{\mathrm{raw}}, \Delta t^{\mathrm{raw}}, b_{\mathrm{gain}}, B_{\mathrm{in}}$
 2: **Initialization (Fourier-anchored)**
 3: $\omega \leftarrow \pi\,\sigma(\theta), \quad r \leftarrow \sigma(\rho), \quad A \leftarrow \mathrm{softplus}(A^{\mathrm{raw}}), \quad \Delta t \leftarrow \mathrm{softplus}(\Delta t^{\mathrm{raw}}) + \varepsilon$     ▷
    bounded/stable reparameterizations
 4: Place $\omega$ on a DFT-like grid over $(0,\pi)$;   set $r \approx 1$;   set $b_{\mathrm{gain}} \leftarrow 1$;   optionally neutralize BE
    correction by $A^{\mathrm{raw}} \leftarrow 0, \ \Delta t^{\mathrm{raw}} \ll 0$     ▷ Fourier-like start
 5: $c \leftarrow \cos\omega, \ s \leftarrow \sin\omega; \quad y \leftarrow 0, \ z \leftarrow 0$     ▷ precompute trigs; zero state
 6: **Streaming update (for $t = 0..L-1$)**
 7: **for** $t = 0$ to $L-1$ **do**
 8:     $y' \leftarrow y \odot c \ - \ z \odot s$     ▷ rotation (cosine/sine mix)
 9:     $z' \leftarrow y \odot s \ + \ z \odot c$     ▷ rotation (quadrature advance)
 10:    $y^\star \leftarrow r \odot y' \ + \ b_{\mathrm{gain}} \odot x[t]$     ▷ damping & input pre-gain
 11:    $z^\star \leftarrow r \odot z'$     ▷ damping
 12:    $\mathrm{inv} \leftarrow \big(1 + (\Delta t \odot \Delta t) \odot A\big)^{-1}$     ▷ Backward-Euler correction (per-mode scalar)
 13:    $\mathrm{inj} \leftarrow x[t] \odot B_{\mathrm{in}}$     ▷ input injection into the state
 14:    $y \leftarrow \big(y^\star + \Delta t \odot z^\star + (\Delta t \odot \Delta t) \odot \mathrm{inj}\big) \odot \mathrm{inv}$     ▷ stabilized $y$ update
 15:    $z \leftarrow z^\star - \Delta t \odot (A \odot y) + \Delta t \odot \mathrm{inj}$     ▷ stabilized $z$ update
 16: **end for**
**Output:** $o_t \leftarrow \big(y \odot \cos\phi + z \odot \sin\phi\big) + j\big(z \odot \cos\phi - y \odot \sin\phi\big)$     ▷ complex readout

---

(Figure 2). The construction mirrors a discretized, stabilized state-space with explicit frequency control, enabling both DFT-like initialization and task-driven adaptation. The overall computation algorithm is outlined in Algorithm 1.

**State and inputs.** Each input feature is decomposed across $K$ modal oscillators that together act as a learnable filter bank over time. Concretely, for each input feature $d \in \{1, \ldots, D_{\mathrm{in}}\}$ and mode $k \in \{1, \ldots, K\}$, we maintain a two-dimensional *quadrature* state that tracks cosine/sine coordinates of a rotating phasor driven by the current input:

$$\boldsymbol{s}_t^{(d,k)} \ = \ \begin{bmatrix} y_t^{(d,k)} \\ z_t^{(d,k)} \end{bmatrix} \in \mathbb{R}^2, \qquad x_t^{(d)} \in \mathbb{R}.$$

Here, $y$ and $z$ capture in-phase and quadrature components, respectively, providing a minimal real-valued representation of complex modulation.

**Reparameterizations (learnable).** To ensure stability, we express the mode parameters through bounded or nonnegative reparameterizations. Frequencies $\omega_k$ are confined to $(0, \pi)$ (positive half frequencies with DC and Nyquist components), radii $r_k$ to $(0, 1)$ for damping, and step/viscosity parameters are kept positive via softplus. These definitions control the spectral location, decay, and numerical conditioning of each mode:

$$\omega_k = \pi\,\sigma(\theta_k) \in (0, \pi), \qquad\qquad r_k = \sigma(\rho_k) \in (0, 1),$$
$$A_k = \mathrm{softplus}(A_k^{\mathrm{raw}}) \geq 0, \qquad\qquad \Delta t = \mathrm{softplus}(\Delta t^{\mathrm{raw}}) > 0,$$

with $\sigma$ the logistic sigmoid and $\alpha_k = \big(1 + (\Delta t)^2 A_k\big)^{-1}$. Intuitively, $\omega_k$ sets the mode's center frequency, $r_k$ its per-step attenuation, and $(A_k, \Delta t)$ tune the Backward-Euler stabilization.

**Rotation and damping.** Each modal state advances by a rotation at $\omega_k$ combined with damping $r_k$. The rotation matrix

$$R(\omega_k) \ = \ \begin{bmatrix} \cos\omega_k & -\sin\omega_k \\ \sin\omega_k & \cos\omega_k \end{bmatrix}.$$

encodes the ideal circular motion on the $(y, z)$ plane; multiplying by $r_k$ shrinks the radius, implementing an exponentially decaying sinusoid. This realizes a stable, frequency-selective oscillator per mode.

**Linear recurrence (with Backward-Euler stabilization).** We couple the rotating state to the input through a linear, per-mode recurrence. Backward-Euler stabilization yields a numerically robust discretization preserving stability. The update is linear in the state and input,

$$\boldsymbol{s}_{t+1}^{(d,k)} \;=\; \underbrace{\tilde{A}_k(\omega_k, r_k, A_k, \Delta t)}_{\text{learnable}} \boldsymbol{s}_t^{(d,k)} \;+\; \underbrace{\tilde{B}_k^{(d)}\big(b_{\text{gain}}^{(d,k)}, B_{\text{in}}^{(d,k)}, A_k, \Delta t\big)}_{\text{learnable}} \, x_t^{(d)}.$$

Explicit coefficients are

$$P_y = \alpha_k \, r_k \big( \cos \omega_k + \Delta t \, \sin \omega_k \big), \qquad Q_y = \alpha_k \, r_k \big( \Delta t \, \cos \omega_k - \sin \omega_k \big),$$
$$P_z = r_k \sin \omega_k - \Delta t \, A_k \, P_y, \qquad Q_z = r_k \cos \omega_k - \Delta t \, A_k \, Q_y,$$

so the state transition takes the form

$$\tilde{A}_k \;=\; \begin{bmatrix} P_y & Q_y \\ P_z & Q_z \end{bmatrix}$$

The input provides per-feature, per-mode control over how energy is injected into the oscillator,

$$\tilde{B}_k^{(d)} \;=\; \begin{bmatrix} \alpha_k \Big( b_{\text{gain}}^{(d,k)} + (\Delta t)^2 B_{\text{in}}^{(d,k)} \Big) \\ \Delta t \, \alpha_k \Big( B_{\text{in}}^{(d,k)} - A_k \, b_{\text{gain}}^{(d,k)} \Big) \end{bmatrix} \in \mathbb{R}^2.$$

Together, $(\tilde{A}_k, \tilde{B}_k^{(d)})$ implement a damped, driven oscillator that acts as a narrowband, learnable filter centered at $\omega_k$.

**Readout ($C$-matrix): complex bins.** After advancing the state, we can project each mode onto a phase-aligned complex axis. The complex readout aligns phases by a fixed offset $\phi_k$ so that each mode matches a DFT bin at a reference length: Let $\phi_k = \omega_k \, (L_{\text{ref}} - 1)$. Then the real/imag parts are

$$\Re \hat{X}_t^{(d,k)} \;=\; y_t^{(d,k)} \cos \phi_k + z_t^{(d,k)} \sin \phi_k, \, \Im \hat{X}_t^{(d,k)} \;=\; -y_t^{(d,k)} \sin \phi_k + z_t^{(d,k)} \cos \phi_k.$$

**Initialization as Half DFT.** To connect with classical Fourier analysis and to facilitate faster convergence, we initialize the modes to mimic a block DFT. Specifically, we place $\omega_k$ on a DFT-like grid, set $r_k \approx 1$ (nearly undamped), choose readout phases via $\phi_k$, and neutralize Backward-Euler corrections. From zero initial state, output at step $L_{\text{ref}}$ therefore matches a DFT across the window, after which learning adjusts frequencies, dampings, and gains to yield task-optimal, frequency-aware filtering while retaining linear-time recurrence and streaming capability.

**Real vs. complex spectra and redundancy.** For real-valued inputs, conjugate symmetry implies that all information is contained in the positive half of the spectral parameters; we therefore learn and compute only the positive half and, when needed, recover the full set by conjugate mirroring. In contrast, complex baseband (I-Q) signals do not exhibit this redundancy. Let $x_{\text{IQ}}[n] = x_I[n] + j \, x_Q[n]$ with $x_I, x_Q \in \mathbb{R}$. We first estimate the positive-half spectral parameters $\tilde{S}_I[k], \tilde{S}_Q[k]$ for $k = 0, \ldots, N/2$. Each channel's full parameters are completed by conjugation on the negative indices, and the complex spectral parameters are then assembled as

$$S_{\text{IQ}}[k] \;=\; S_I[k] \;+\; j \, S_Q[k], \qquad k = 0, \ldots, N-1.$$

# 4 EXPERIMENTS AND RESULTS

## 4.1 RADAR OBJECT DETECTION PIPELINE

### 4.1.1 DATASET

**RADIal (HD automotive radar).** RADIal [Rebut et al. (2022b)] is a raw high-definition FMCW automotive radar dataset comprising **91** driving sequences (∼1-4 minutes each; ∼2 hours total), with approximately **25,000** synchronized frames, of which **8,252** are annotated for *vehicle* detection and *drivable-area* segmentation on the range-azimuth grid. The radar sensor is Doppler division multiplexed (DDM) with **12 transmit** and **16 receive** antennas (**192** virtual antennas). Labels target vehicles only for detection (range-angle maps) and free-space for segmentation. We adopt the official **70/15/15** train/validation/test split.

### 4.1.2 METHODOLOGY FOR FSSM INTEGRATION

Let the dechirped I-Q ADC tensor for one frame be

$$\mathbf{x} \in \mathbb{R}^{2 \times N_s \times N_c \times M},$$

stacking in-phase and quadrature along the first axis (2), with $N_s$ fast-time samples per chirp, $N_c$ chirps (slow time) per frame, and $M$ virtual array channels.

**FSSM along samples (range).** We denote by $\mathcal{F}^{(s)}$ the FSSM applied *along the sample axis* (fast time, index $n = 0, \ldots, N_s - 1$) independently for each chirp and antenna. This block extracts $K_r$ range spectral parameters per location:

$$\mathbf{S}^{(r)} = \mathcal{F}^{(s)}(\mathbf{x}; \text{axis} = \text{samples}) \in \mathbb{R}^{2 \times K_r \times N_c \times M}, \tag{1}$$

encoding complex range features as in Section 3.1. Intuitively, $\mathcal{F}^{(s)}$ replaces the fixed FFT over fast time with learnable range bins.

**FSSM along chirps (Doppler).** Conditioned on range, we next apply a second FSSM $\mathcal{F}^{(c)}$ *along the chirp axis* (slow time, index $\ell = 0, \ldots, N_c - 1$) to obtain $K_d$ Doppler spectral parameters per (range, antenna):

$$\mathbf{S}^{(rd)} = \mathcal{F}^{(c)}(\mathbf{S}^{(r)}; \text{axis} = \text{chirps}) \in \mathbb{R}^{2 \times K_r \times K_d \times M}, \tag{2}$$

which plays the role of a learnable Doppler transform over slow time.

**Angle projection and decoding.** We project the array dimension into $K_\theta$ discrete azimuth bins using a learnable angle head $\mathcal{A}_\varphi$ (e.g., 1D convolutions across $M$ channels that emulate/augment beamforming):

$$\mathbf{Z} = \mathcal{A}_\varphi(\mathbf{S}^{(rd)}) \in \mathbb{R}^{K_r \times K_d \times K_\theta}. \tag{3}$$

A convolutional decoder $\mathcal{D}_\psi$ (a stack of $3{\times}3 / 1{\times}1$ convolutions with interleaved nonlinearity and normalization, as in FFTRadNet [Rebut et al. (2022b)] maps $\mathbf{Z}$ to task heads for detection and segmentation. Concretely, we produce range-angle detection heatmaps and semantic masks:

$$\left(\hat{\mathbf{Y}}^{\text{det}}, \hat{\mathbf{Y}}^{\text{seg}}\right) = \mathcal{D}_\psi(\mathbf{Z}), \qquad \hat{\mathbf{Y}}^{\text{det}} \in [0,1]^{K_r \times K_\theta \times C_{\text{det}}}, \quad \hat{\mathbf{Y}}^{\text{seg}} \in [0,1]^{K_r \times K_\theta \times C_{\text{seg}}}. \tag{4}$$

In practice, Doppler can be retained as a conditioning channel within $\mathbf{Z}$ or partially pooled prior to the final heads, following the multi-task design in the RADIal pipeline.

### 4.1.3 TRAINING PROTOCOL

We train the radar models (FSSM-FFTRadNet, FSSM-TFFTRadNet, FourierNet-FFTRadNet, etc.) with batch size of 4 using Adam optimizer with an initial learning rate $1 \times 10^{-4}$ and a step scheduler (decay factor $\gamma = 0.9$ every 10 epochs) for a total of 100 epochs. The multi-task objective follows PixorLoss with focal classification and Smooth L1 regression. Loss weights are $[1, 100, 100]$ for classification, regression, and segmentation, respectively.

### 4.1.4 RADAR OBJECT DETECTION RESULTS

Table 1 reports segmentation and detection performance across state-of-the-art convolutional and attention-based architectures.

Within *convolutional backbones*, our **FSSM-FFTRadNet** achieves the strongest detection performance (**F1 0.98**, **mAP 0.98**, **mAR 0.99**), outperforming FFT-based models (FFT-RadNet), learnable-DFT models (ADCNet), and the **RFMamba-TFFTRadNet** baseline. Notably, FSSM reaches these gains without increasing computational cost, maintaining nearly the same GMACs and parameter count as FFT-RadNet, while also reducing azimuth error to **0.10°**.

The RFMamba-TFFTRadNet baseline replaces our front-end with RFMamba [Zhang et al. (2025)] (while keeping the same decoder) and yields **F1 0.83**, **mAP 0.84**, and **mAR 0.83**. This highlights the benefit of learning both center frequencies and bandwidths directly from ADC I/Q, rather than learning frequency based features after FFT.

Table 1: Overall segmentation and detection performance on **RADIal** [Rebut et al. (2022a)] grouped by architecture class.

| Class | Method | Model | Seg. | Detection | | | | | Computational Metrics* | | |
|---|---|---|---|---|---|---|---|---|---|---|---|
| | | | mIoU | F1 | mAP | mAR | RE-m | AE-° | GMAC | Param-M | Latency-ms |
| | FFT | Pixor (PC) [Yang et al. (2018)] | — | 0.48 | 0.96 | 0.32 | 0.17 | 0.25 | — | — | — |
| | FFT | Pixor (RA) [Yang et al. (2018)] | — | 0.87 | 0.96 | 0.82 | **0.10** | 0.20 | — | — | — |
| | FFT | PolarNet [Nowruzi et al. (2020)] | 0.61 | — | — | — | — | — | — | — | — |
| | FFT | FFT-RadNet [Rebut et al. (2022a)] | 0.74 | 0.87 | 0.97 | 0.82 | 0.11 | 0.17 | 146.58 | 3.79 | 47.71 |
| | FFT | FFT-RadUNet[a] | 0.75 | 0.80 | 0.83 | 0.77 | 0.16 | 0.09 | 134.40 | 18.48 | 44.92 |
| | L-DFT | ADC UNet [Zhang et al. (2023)] | 0.77 | 0.85 | 0.88 | 0.82 | 0.18 | 0.11 | — | 17.50 | 8.18** |
| Conv. | L-DFT | ADC UNet (NPT) [Zhang et al. (2023)] | 0.73 | 0.80 | 0.83 | 0.77 | 0.19 | 0.10 | — | — | — |
| | L-DFT | ADCNet [Zhang et al. (2023)] | 0.79 | 0.89 | 0.93 | 0.86 | 0.13 | 0.11 | — | 2.50 | 18.13** |
| | L-DFT | FourierNet-FFT-RadUNet[b] | 0.78 | 0.86 | 0.84 | 0.87 | 0.16 | 0.11 | 134.41 | 19.13 | 48.73 |
| | L-DFT | FourierNet-FFT-RadNet[c] | 0.79 | 0.88 | 0.87 | 0.89 | 0.14 | 0.12 | 150.67 | 4.45 | 51.66 |
| | RF-SSM | RFMamba–TFFTRadNet [Zhang et al. (2025)] | 0.75 | 0.83 | 0.84 | 0.83 | 0.18 | 0.20 | 268.76 | 6.79 | — |
| *Best (Conv.)* | **FSSM** | **FSSM-FFTRadNet[d]** | **0.81** | **0.98** | **0.98** | **0.99** | 0.12 | **0.10** | 147.49 | 3.84 | 61.27 |
| Attn. | FFT | TransRadar [Dalbah et al. (2024)] | 0.81 | 0.98 | 0.97 | 0.98 | 0.11 | 0.10 | — | 3.4 | — |
| | L-DFT | TFFTRADNet [Giroux et al. (2023)] | 0.79 | 0.87 | 0.88 | 0.87 | 0.16 | 0.13 | 104.34 | 10.29 | 52.90 |
| *Best (Attn.)* | **FSSM** | **FSSM-TFFTRadNet[e]** | **0.85** | **0.98** | **0.98** | 0.97 | **0.10** | **0.07** | 101.15 | 9.69 | 59.88 |

FFT – Fast Fourier Transform; L-DFT – Learned Discrete Fourier Transform; RF-SSM – RF Mamba-style state-space encoder; FSSM – (ours); Attn. – attention-based detection models; Conv. – convolution-based detection models.
[a,b,c,d,e] [a] FFT-RadNet [Rebut et al. (2022a)] + UNet [Ronneberger et al. (2015)]; FourierNet [Zhao et al. (2023)] DFT fed to [b] FFT-RadUNet, [c] FFT-RadNet; our FSSM preprocessing with [d] [Rebut et al. (2022a)], [e][Giroux et al. (2023)]. RFMamba–TFFTRadNet is our implementation of the RF-Mamba encoder [Zhang et al. (2025)] plugged into the TFFTRadNet decoder.
*Runtime characterization for **segmentation + detection** on an NVIDIA RTX 4060 mobile GPU unless otherwise noted. **Reported by Zhang et al. (2023) on an NVIDIA RTX 3090 GPU.

In the *attention-based* family, **FSSM-TFFTRadNet** sets a new state of the art with **F1 0.98** and **mIoU 0.85**, improving upon TransRadar while reducing azimuth error to **0.07°**. Across both architectural classes, the FSSM front-end consistently delivers superior detection accuracy under comparable or lower computational budgets.

### 4.1.5 ROBUSTNESS ANALYSIS

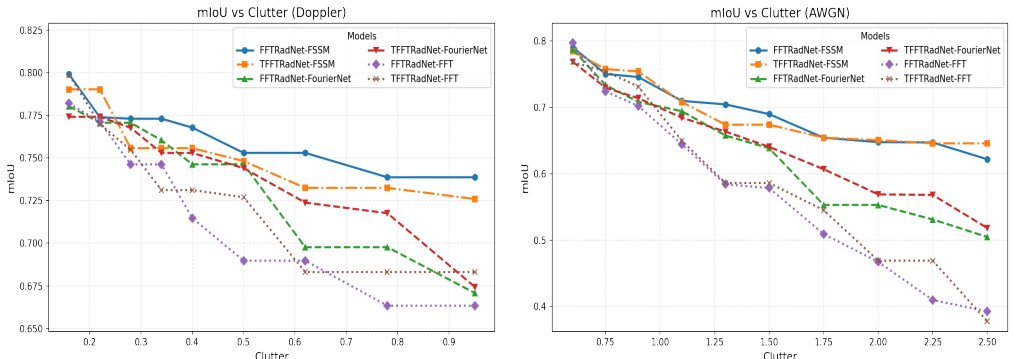

Figure 3: Noise ablation on RADIal under (a) spatial-temporal clutter (Gaussian smoothing in Doppler) and (b) additive white Gaussian noise (AWGN) at varying SNR. FSSM (ours) consistently outperforms learnable Fourier and fixed FFT front-ends across perturbation levels.

Under additive synthetic perturbations, radar segmentation performance degrades as clutter increases. For AWGN (Figure 3b), lowering SNR reduces mIoU for all models, with FSSM least affected (most robust), FourierNet showing moderate loss, and FFT degrading the most. For spatial-temporal clutter, as the Gaussian smoothing parameter $\sigma$ increases (Figure 3a), the same ranking holds: FSSM > FourierNet > FFT; with FFT exhibiting the steepest drop. (See Section A.6 for details of the radar clutter experiments.)

## 4.2 Audio Keyword Detection Pipeline

### 4.2.1 Experimental Audio Architecture for Integration

Let the mono audio waveform be $\mathbf{x} \in \mathbb{R}^N$ sampled at $F_s$ Hz. We use a fixed window of $L$ samples and hop $H$ ($H < L$). The number of frames is

$$T = 1 + \left\lfloor \frac{N - L}{H} \right\rfloor,$$

with frame indices $t = 0, \ldots, T-1$ and sample indices $n = 0, \ldots, L-1$.

**FSSM per window (magnitude only).** For each windowed segment $\mathbf{x}_t[n] = x[tH + n] \cdot w[n]$ with an analysis window $w[\cdot]$, we apply the FSSM along the *sample axis* (as in Section 3.1) and retain only the magnitude readout. Denote this operator by $\mathcal{F}_{\Theta}^{(s)}$ with $K_a$ spectral bins:

$$\mathbf{s}_t = \mathcal{F}_{\Theta}^{(s)}\big(\mathbf{x}_t; \text{axis} = \text{samples}, \text{out} = \text{mag}\big) \in \mathbb{R}^{K_a}, \qquad \mathbf{S} = \big[\mathbf{s}_0; \ldots; \mathbf{s}_{T-1}\big] \in \mathbb{R}^{T \times K_a}. \tag{5}$$

Thus $\mathbf{S}$ is a learnable spectrogram with $T$ time steps and $K_a$ frequency features (bins) per step.

**Bidirectional Mamba over frames.** We treat the rows of $\mathbf{S}$ as a sequence of $T$ tokens with feature dimension $K_a$ and process it with $B$ stacked bidirectional Mamba [Gu & Dao (2023)] blocks $\mathcal{M}_{\Gamma}^{(\leftrightarrow)}$:

$$\mathbf{H}^{(0)} = \mathbf{S}, \qquad \mathbf{H}^{(b)} = \mathcal{M}_{\Gamma_b}^{(\leftrightarrow)}\big(\mathbf{H}^{(b-1)}\big) \in \mathbb{R}^{T \times K_a}, \ b = 1, \ldots, B, \tag{6}$$

yielding $\mathbf{H} = \mathbf{H}^{(B)}$ with the same shape, where forward/backward selective state updates aggregate context across frames.

**Temporal collapse and classification.** We temporally collapse the $T$ frames with a 1D convolution along the time axis to obtain a single feature vector whose length equals the number of spectral bins $K_a$:

$$\mathbf{z} = \text{Conv1D}_t(\mathbf{H}) \in \mathbb{R}^{K_a}. \tag{7}$$

Finally, a multilayer perceptron $\mathcal{G}_\eta$ maps $\mathbf{z}$ to class logits for $C$ labels:

$$\hat{\mathbf{y}} = \mathcal{G}_\eta(\mathbf{z}) \in \mathbb{R}^C. \tag{8}$$

In summary, the pipeline replaces a fixed STFT with a learnable, magnitude-only FSSM per window, models inter-frame dynamics via bidirectional Mamba, and performs temporal pooling with a 1D convolution before classification.

### 4.2.2 Training Protocol and Evaluation

Table 2: Front-end comparison across Speech Commands V2 [Warden (2018)](Top-1 accuracy) and AudioSet [Gemmeke et al. (2017)](balanced mAP).

(a) Speech Commands V2 (Bi-Mamba backend [Gu & Dao (2023)]

| Front-end | Top-1 Acc. (%) |
|---|---|
| None (raw waveform) | 90.54 |
| FFT / STFT | 93.23 |
| FourierNet-style learned DFT | 93.41 |
| SincNet front-end | 94.73 |
| LEAF (learnable filterbank) | 94.86 |
| **FSSM (ours)** | **97.16** |

(b) AudioSet(AST backend [Gong et al. (2021)])

| Front-end | Balanced mAP |
|---|---|
| FFT spectrograms | 0.340 |
| **FSSM (ours)** | **0.365** |

We train the audio models (FFT/FourierNet/FSSM encoders followed by 8 layers of bidirectional Mamba blocks; input dimension $= 200$, sequence length $= 99$) with batch size $= 128$. We use Adam optimizer at an initial learning rate $2.5 \times 10^{-4}$ for 50 epochs, and a step scheduler that starts

at epoch 5 and decays the learning rate by 0.85 every epoch thereafter (no warmup). The objective is cross-entropy over one-hot targets, and accuracy is the primary evaluation metric.

Table 2 shows that, under an identical Bi-Mamba backend and training protocol, FSSM consistently outperforms both fixed (FFT/STFT) and learnable spectral front-ends (FourierNet, SincNet, LEAF) on Speech Commands V2 and AudioSet.

### 4.2.3 ROBUSTNESS EXPERIMENTS IN AUDIO KEYWORD DETECTION

We evaluate robustness on Speech Commands V2 [Warden (2018)] using the standard 35-class split and the official train/val/test partitions. Each model shares the same downstream classifier; only the front-end differs: FFT (fixed STFT features), FourierNet (learnable Fourier basis), and FSSM (ours; state-space spectral module). We inject synthetic additive perturbations at the waveform level and sweep the signal-to-noise ratio (SNR) over a broad range. Figure 4 shows FSSM's performance degrades more slowly for decreasing SNRs compared to FFT and learnable DFT methods. Under heavy perturbations at 0 dB, our approach outperforms other methods by 4-20%, showing its robustness under noise augmentation.

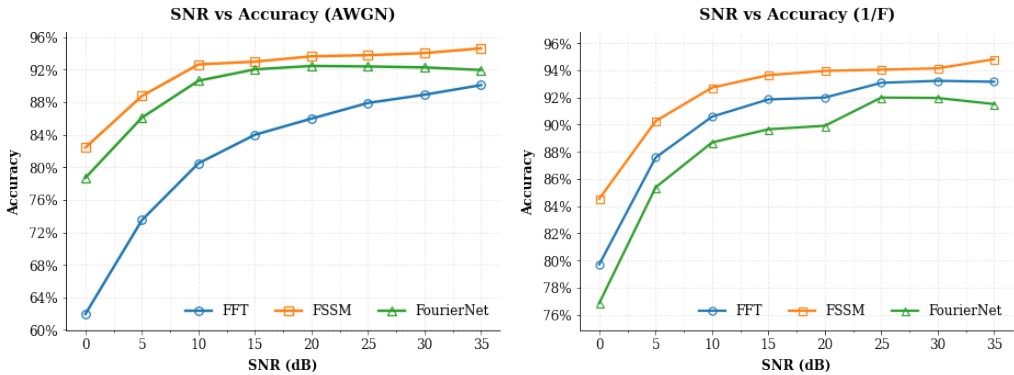

Figure 4: Noise ablation on Speech Commands V2 [Warden (2018)] under (a) AWGN and (b) pink noise. FSSM (ours) consistently outperforms learnable Fourier and fixed FFT front-ends across SNR levels.

### 4.2.4 ABLATIONS AND VISUALIZATION ON FREQUENCY SELECTIVITY

**Frequency Selectivity.** In our formulation, a band corresponds to a single FSSM mode, i.e., a complex narrowband filter with a learnable center frequency $\omega_k$ and bandwidth parameter $\alpha_k$. In contrast, an FFT band corresponds to a fixed frequency bin at $\omega_k = \frac{2\pi k}{N}$ associated with the $k$-th Fourier coefficient [Oppenheim (1999)]. This distinction means FSSM can reshape both where and how broadly it allocates spectral coverage, enabling true task-adaptive frequency selectivity.

To make this effect explicit, we introduce a **frozen-band ablation**. We initialize all FSSM modes to match a standard Fourier grid, and then compare: (i) FFT/STFT with fixed bins, (ii) FSSM with frozen Fourier bands (bin centers and bandwidths fixed), and (iii) FSSM with fully learned bands (ours). This isolates whether improvements arise simply from reparameterizing an FFT, or from learning the band structure itself.

Table 3: Frozen-band ablation showing the role of learned frequency selectivity.

| Front-end variant | RADIal F1 | RADIal mAP | SC-V2 Top-1 |
|---|---|---|---|
| FFT / STFT | 0.872 | 0.971 | 93.230 |
| FSSM (frozen Fourier bands) | 0.883 | 0.978 | 93.621 |
| **FSSM (learned bands, ours)** | **0.982** | **0.983** | **97.160** |

Two trends consistently emerge. First, on **RADIal**, freezing the bands yields only marginal gains over FFT (0.872→0.883 F1), whereas fully learned FSSM provides a large improvement

(0.982 F1). Second, on **Speech Commands V2**, the frozen-band model offers only a small boost (93.23%→93.62%), but the learned-band model achieves 97.16%. These results show that FSSM's benefits do not arise from a trivial Fourier-like initialization, but from its ability to *learn which frequencies to emphasize and how wide each band should be*.

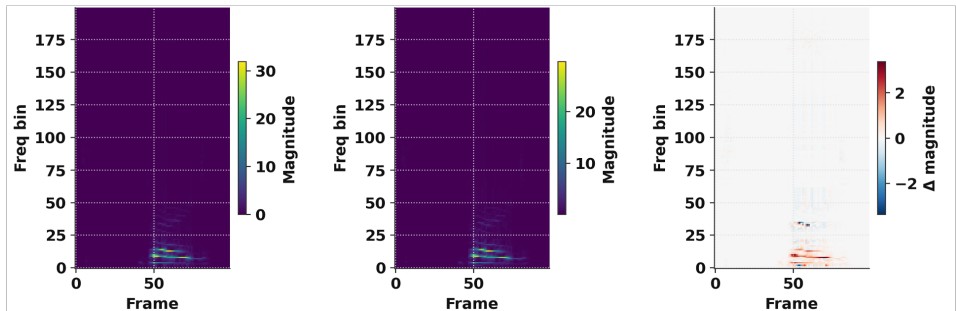

Figure 5: **Spectral difference of FFT vs. FSSM on speech spectrograms.** For a 1 s audio sample with 25 ms windows and 15 ms hop, we visualize the FFT (left) and the FSSM representation (middle). The difference heatmap (right) highlights selective frequency gain adjustments in regions of interest (higher signal power).

Qualitatively for audio, we show in Figure 5 FFT spectrograms, FSSM spectrograms, and FFT-FSSM difference maps across diverse utterances. FSSM consistently sharpens formant and harmonic regions while suppressing background energy in less informative bands [Ravanelli & Bengio (2018); Zeghidour et al. (2021)]. Together, the frozen-band ablation and these visualizations demonstrate that FSSM learns a genuinely task-adaptive spectral basis rather than acting as a reparameterized FFT.

**Effect of Learnable Damping.** We ablate the role of the damping parameter by fixing $r_k = 1$ for all modes, thereby removing explicit bandwidth control and reducing each mode to an undamped sinusoid. This constraint leads to a clear degradation in performance, particularly in low-SNR or noisy regimes: accuracy on Speech Commands drops by up to 2.5%, and mIoU/F1 on RADIal decreases by approximately 3.8%. These results highlight the importance of allowing the model to tune the effective window-length of each mode, which directly governs spectral leakage and side-lobe behavior [Harris (2005); Oppenheim (1999)].

## 5 CONCLUSION

We presented a Fourier-initialized, frequency-selective state-space module (FSSM) that learns stable, causal, band-selective kernels as a spectral front-end for time-series data. Trained end-to-end, the module provides task-adapted spectral weights while retaining linear-time, streamable inference. Integrated into radar perception pipelines, FSSM attains state-of-the-art range-azimuth detection and free-space segmentation on RADIal. In speech commands, replacing fixed STFT and prior learnable Fourier blocks with a magnitude-only FSSM front-end followed by lightweight sequence modeling yields consistent accuracy gains and superior robustness under additive perturbations. Visual analyses further indicate that FSSM learns spectral representations that optimize spectra to concentrate energy on task-relevant bands.

**Future directions.** Future work will explore: (i) input-dependent spectral weight selection to dynamically retune frequency emphasis per instance or spectral adaptation from long-sequence memory thus leveraging streaming state to modulate spectra over time; (ii) cross-mode feature sharing to couple and regularize neighboring frequency bands; (iii) cross-channel feature sharing to better exploit structure across antennas/sensors or microphone-array inputs; -toward a generalizable foundation model for spectral representations of time-series signals like radar, audio and RF signal processing domains.

**Disclosure of use of Large Language Models** Text generation for polishing writing and some of the research content discovery was done with the help of ChatGPT 5.0.

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

# A APPENDIX

## A.1 RADAR OBJECT DETECTION BACKGROUND

**Fourier foundations for FMCW radar.** Frequency-modulated continuous-wave (FMCW) radars transmit linear chirps $s_{\text{tx}}(t) = \exp\big(j2\pi(f_c t + \frac{S}{2}t^2)\big)$ with carrier $f_c$ and slope $S$. After dechirping, a point target at range $R$ yields, within one chirp, a (nearly) single-tone *beat* with "fast-time" frequency

$$f_r \approx \frac{2SR}{c}, \tag{9}$$

where $c$ is the speed of light. Stacking $L$ chirps forms a slow-time sequence whose inter-chirp phase advance encodes the (radial) Doppler frequency

$$f_d = \frac{2v}{\lambda}, \qquad \lambda = \frac{c}{f_c}, \tag{10}$$

with $v$ the target radial velocity. With a uniform linear (virtual) array of $M$ elements and spacing $d$, the per-element phase shift for azimuth $\theta$ is $a_m(\theta) = \exp\big(j2\pi m \frac{d}{\lambda}\sin\theta\big)$, and an angular FFT (or beamformer) across channels localizes direction:

$$\hat{\theta} \in \arg\max_{\theta} \left\| \sum_{m=0}^{M-1} x_m\, e^{-j2\pi m \frac{d}{\lambda}\sin\theta} \right\|. \tag{11}$$

Consequently, range, Doppler, and angle are estimated by FFTs over fast-time, slow-time, and channel axes, producing standard 2D/3D "radar cubes" such as range-Doppler (RD), range-angle (RA), and range-Doppler-angle (RDA).

**Learning on radar spectra and cubes.** Early deep architectures established that RD/RDA tensors could be treated as images and processed end-to-end. Major et al. (2019) demonstrated vehicle detection directly on Range-Azimuth-Doppler tensors with 3D CNNs, showing that joint spectral-spatial features learned from the cube surpass classical peak-based pipelines in cluttered scenes. RODNet introduced cross-modal supervision-training radar heatmap detectors with labels derived from fused camera-radar 3D localization-to mitigate annotation noise, while exploiting multi-frame context to stabilize predictions across time [Wang et al. (2021)]. View-multiplexing strategies such as RAMP-CNN learn separate encoders on RD/RA/AD slices and fuse them, capturing complementary statistics across frequency, range, and aperture at the cost of additional alignment and compute [Gao et al. (2020)].

With the advent of high-definition arrays and public raw-signal corpora, methods began to *learn* the spectral front-end itself. On the RADIal dataset, FFTRadNet attached a learnable Fourier stage to a dense CNN backbone, enabling multi-task segmentation/detection directly from ADC while retaining the interpretability and efficiency of FFT-like structure [Rebut et al. (2022b)]. T-FFTRadNet replaced the convolutional trunk with a Swin-style transformer to aggregate long-range spectral context across tokens, improving sensitivity to weak or far-range returns albeit with higher memory footprint [Giroux et al. (2023)]. In parallel, ADCNet showed that knowledge-distillation from processed representations can guide a student that operates purely on raw waveforms, narrowing the optimization gap of end-to-end training on ADC streams [Yang et al. (2023)].

Beyond a single sensor or static frames, system- and sequence-level designs further advance robustness. Radatron fused a cascaded pair of MIMO radars with complementary fields-of-view via a multi-resolution feature pyramid, boosting distant and small-object performance where single-sensor RD maps are resolution-limited [Madani et al. (2022)]. Recurrent CNNs over RD videos (e.g., ConvLSTM backbones) leverage slow-time coherence to suppress transient clutter and emphasize micro-Doppler, thereby stabilizing online detection under low SNR and ego-motion [Decourt et al. (2024)].

## A.2 Speech Command Recognition on Spectrograms

**STFT and spectrograms.** Speech is quasi-stationary over short windows (typically 20-40 ms). The short-time Fourier transform applies a window $w$ centered at frame index $t$,

$$X(t, \omega) = \sum_{\tau} x[\tau] \, w[\tau - t] \, e^{-j\omega\tau}, \tag{12}$$

and advances by a hop $H$, yielding overlapping frames when $H < |w|$. The spectrogram $|X(t, \omega)|^2$ (frequently log-scaled and projected to mel bands) is thus a time-frequency image: each column summarizes local frequency content within a short frame, and the horizontal evolution encodes onsets, formants, and harmonics over the utterance.

**Image-style and sequential models.** Compact convolutional designs such as MatchboxNet apply time-channel separable convolutions to log-mel spectrograms, attaining on-device keyword spotting with very low latency and parameter counts by emphasizing local time-frequency structure [Majumdar & Ginsburg (2020)]. Transformer-only encoders like AST tokenize spectrogram patches and learn global time-frequency dependencies via self-attention, which improves transfer and robustness on large-scale audio tagging and speech-command benchmarks when sufficient pretraining is available [Gong et al. (2021)]. For full ASR, Conformer interleaves multi-head attention with local 1D convolutions, marrying global context with phonetic locality to reach state-of-the-art word error rates on LibriSpeech [Gulati et al. (2020)]. Orthogonally, self-supervised front-ends (e.g., wav2vec 2.0) learn latent acoustic features from raw audio that can be fine-tuned for downstream modeling; such encoders can precede spectrogram-based decoders or replace explicit spectrograms entirely, depending on compute and data regimes [Baevski et al. (2020)]. Across these approaches, the central tension is between exploiting spectrogram locality (CNNs) and capturing long-range structure (Transformers and SSL), with hybrids offering a practical balance for command-level tasks.

## A.3 Additional Background on SSM Initialization

**From structured to selective dynamics.** S4 introduced a carefully initialized, structured state matrix $A$ (HiPPO-based) enabling fast frequency-domain convolution and stable long-context training, establishing SOTA on long-range benchmarks [Gu et al. (2021)]. DSS simplified the parameterization by using a diagonal $A$, showing that much of S4's power persists with independent first-order filters per channel and faster training [Gupta et al. (2022)]. S5 coupled channels via a single multi-input/multi-output SSM per layer and leveraged parallel scans, improving utilization and accuracy while retaining linear scalability [Smith et al. (2022)]. Mamba transformed SSMs from fixed linear systems into *input-selective* ones by generating $(A, B)$ as functions of the current token, which integrates content-dependent gating with linear-time inference and scales competitively to Transformer quality in language, audio, and beyond [Gu & Dao (2023)]. Complementary frequency-aware variants inject spectral context into SSM processing: RF-Mamba merges adaptive frequency features with time-domain dynamics for RF/radar sensing, and Vim-F augments visual SSMs with FFT-derived global context, both demonstrating gains in domains where frequency structure is predictive [Zhang et al. (2025; 2024)].

**Bases, orthogonality, and Fourier anchoring.** Different SSM parameterizations implicitly select different basis functions for representing long histories: HiPPO in S4 emphasizes polynomial memory; diagonal DSS behaves like a bank of decoupled IIR filters; coupled S5 mixes channels through a shared dynamical system. In practice, anchoring modes to frequency-via evenly spaced rotations with lightly damped radii-provides a Fourier-like initialization that approximates a learnable filter bank. Sequential scanning then adapts per-mode gains, phases, and dampings to track nonstationary spectral components over time, complementing block DFTs with instance- and time-dependent selectivity.

## A.4 Analytical Insight Into the Fourier-Selective State-Space Module

FSSM's core object is analytically simple and closely related to classical spectral analysis: each FSSM mode has impulse response of a damped complex sinusoid.

$$h_k[n] \propto r_k^n e^{j\omega_k n}$$

This is exactly a narrowband complex IIR filter with learnable center frequency $\omega_k$, and learnable bandwidth / effective window-length controlled by $r_k$. In other words, each mode can be viewed as a narrow band-pass filter in the classical DSP sense: it passes a small range of frequencies around $\omega_k$ while attenuating others.

With DFT-grid initialization and $r_k \approx 1$, summing over a window recovers the usual DFT coefficient for that bin. Thus, FSSM strictly contains FFT/STFT as a special case. This provides a natural explanation for the empirical gains: A block FFT uses fixed, undamped sinusoids and rigid windows, which is known to induce spectral leakage and smearing for off-grid frequencies and non-stationary content [Oppenheim (1999); Harris (2005)].

FSSM preserves the same basic sinusoidal structure but allows each "bin" to move off the grid (learnable $\omega_k$), tighten or widen its effective bandwidth (via $r_k$), and do so differently per task and per domain. In our experiments: on RADIal, FSSM improves F1/mAP while maintaining similar GMACs and parameter counts to FFT / learnable-DFT radar front-ends such as FFTRadNet/FourierNet [Rebut et al. (2022a); Sharma et al. (2024)]. On Speech Commands V2, FSSM consistently outperforms fixed and learnable spectral front-ends (FFT, FourierNet, SincNet, LEAF) under the same Bi-Mamba decoder (see Weakness 3 response), in line with prior work showing the importance of well-shaped, task-adaptive bands [Ravanelli & Bengio (2018); Zeghidour et al. (2021)]. Intuitively, by learning bands that align with the true energy distribution of radar and speech signals, FSSM produces sharper, less smeared spectral features with higher effective SNR. This makes the downstream CNN/SSM/transformer easier to train and explains the consistent F1/mAP/accuracy gains we observe.

## A.5 INCREMENTAL IMPROVEMENTS IN RADAR DETECTION PERFORMANCE

Table 4 reports the mean and standard deviation of key detection metrics (F1, mAP, mAR) for the baseline radar backbones and their FSSM-augmented counterparts. The results show that incorporating the Fourier-Selective module consistently improves end-to-end detection quality without altering the network architecture beyond the spectral front-end.

Table 4: Comparison of mean $\pm$ standard deviation for radar detection metrics, highlighting the incremental gains introduced by the proposed FSSM module.

| Model | F1 | mAP | mAR |
|---|---|---|---|
| FFT-RadNet | $0.872 \pm 0.006$ | $0.971 \pm 0.004$ | $0.824 \pm 0.007$ |
| **FSSM-FFTRadNet (ours)** | $\mathbf{0.982 \pm 0.003}$ | $\mathbf{0.983 \pm 0.002}$ | $\mathbf{0.991 \pm 0.002}$ |
| TFFTRadNet | $0.871 \pm 0.005$ | $0.881 \pm 0.006$ | $0.872 \pm 0.005$ |
| **FSSM-TFFTRadNet (ours)** | $\mathbf{0.981 \pm 0.003}$ | $\mathbf{0.982 \pm 0.003}$ | $\mathbf{0.972 \pm 0.004}$ |

Across both architectural families, the improvements in F1 are substantially larger than their corresponding standard deviations, indicating that the gains are statistically meaningful rather than minor fluctuations. For example, F1 increases from $0.872 \pm 0.006$ in FFT-RadNet [Rebut et al. (2022a)] to $0.982 \pm 0.003$ in FSSM-FFTRadNet, a margin far exceeding the training variance. A similar trend holds for the attention-based TFFTRadNet backbone [Giroux et al. (2023)], where replacing the fixed spectral front-end with FSSM yields consistently higher F1, mAP, and mAR. These results demonstrate that the proposed module provides a robust, repeatable improvement to radar object detection performance while maintaining architectural simplicity and efficiency.

## A.6 MODELING RADAR CLUTTER FOR ML-BASED DETECTION

We study how nuisance environments degrade a learning-based radar detector by injecting one clutter process at a time into the complex baseband I/Q stream and re-evaluating the model. We consider two representative families that capture thermal noise, range-dependent attenuation, and spatial-temporal correlation: (a) additive white Gaussian noise (AWGN), and (b) spatial-temporal (ST) clutter. Each model exposes a single interpretable control parameter that we sweep to obtain stress-response curves.

**(a) Additive White Gaussian Noise (AWGN).** AWGN models receiver/thermal fluctuations with a flat power spectral density. Given a desired $\mathrm{SNR}_{\mathrm{dB}}$, we add zero-mean circular complex Gaussian noise

$$x_{\mathrm{AWGN}}[t] \;=\; x_{\mathrm{orig}}[t] + n[t], \qquad n[t] \sim \mathcal{CN}\big(0, \sigma^2\big), \tag{13}$$

where $\sigma^2$ follows from the average signal power $P_{\mathrm{sig}} = \frac{1}{T}\sum_{t=1}^{T} |x_{\mathrm{orig}}[t]|^2$:

$$\mathrm{SNR}_{\mathrm{dB}} \;=\; 10\log_{10}\!\left(\frac{P_{\mathrm{sig}}}{\sigma^2}\right) \;\implies\; \sigma^2 \;=\; \frac{P_{\mathrm{sig}}}{10^{\mathrm{SNR}_{\mathrm{dB}}/10}}. \tag{14}$$

*Sweep parameter:* $\mathrm{SNR}_{\mathrm{dB}}$. For baseband I/Q, use circular symmetry: $n = \frac{1}{\sqrt{2}}(n_I + jn_Q)$ with $n_I, n_Q \sim \mathcal{N}(0, \sigma^2)$.

**(b) Spatial-Temporal Clutter (ST).** To model correlated multipath/ground clutter and slowly varying interferers, we impose joint smoothing across range, cross-range (or channel), and slow time on the 3-D lattice $(r, c, t)$. With an isotropic Gaussian kernel

$$G_\sigma(\Delta r, \Delta c, \Delta t) \;=\; \frac{1}{(2\pi)^{3/2}\sigma^3} \exp\!\left(-\frac{\Delta r^2 + \Delta c^2 + \Delta t^2}{2\sigma^2}\right), \tag{15}$$

form the convolution

$$x_{\mathrm{ST}}[r, c, t] \;=\; (G_\sigma * x_{\mathrm{orig}})[r, c, t] \;=\; \sum_{\Delta r, \Delta c, \Delta t} G_\sigma(\Delta r, \Delta c, \Delta t)\, x_{\mathrm{orig}}[r-\Delta r,\, c-\Delta c,\, t-\Delta t]. \tag{16}$$

Here $\sigma > 0$ is the correlation length; larger $\sigma$ increases spatial-temporal coherence and can smear targets. *Sweep parameter:* $\sigma$. Implement separably (range/cross-range/time) or via FFTs; normalize $G_\sigma$ to unit sum.

**Evaluation protocol.** For each clutter family, we sweep its control parameter over a fixed grid and measure detector performance (e.g., ROC/AUC, $P_d$-$P_{fa}$) on identically curated scenes:

$$\text{AWGN: } \mathrm{SNR}_{\mathrm{dB}} \in [\text{low, high}], \qquad \text{ST: } \sigma \in [\sigma_{\min},\, \sigma_{\max}].$$

Only one clutter model is active at a time to isolate its effect; compositions approximate severe scenarios but are excluded from this controlled study.

