# OpenReview forum: "FSSM: Frequency-Selective State Space Models for Spectral Representation Learning"
_ICLR.cc/2026/Conference — Submitted to ICLR 2026_

### Official Review · Reviewer_cgvh · 2025-10-21

**Soundness:** 2
**Presentation:** 1
**Contribution:** 2
**Rating:** 4
**Confidence:** 2

**Summary:**

This paper introduces the frequency-selective state space model (FSSM), a spectral representation learning module that augments SSMs with frequency-selective, task-adaptive kernels. It demonstrates the effectiveness of FSSM on two distinct application domains: high-definition automotive radar for object detection and free-space segmentation and speech keyword recognition.

**Strengths:**

* The paper contains a thorough discussion of related works and the proposed method is written rigorously into pseudocode,

**Weaknesses:**

* My major concern is the positioning of this paper. It is advertised for using SSMs to enhance spectral representation learning, but it is barely accessible to either community. For SSM people, section 3 is written in a mysterious way. There are multiple new design choices different from a standard SSM/Mamba and no intuition is given. For spectral learning people, the benefit of SSM is not clearly anticipated and there also lacks enough background provided for understanding the proposed SSM integration.
* There is no analysis of the model in addition to experiments. It is intuitively unclear why the proposed method should outperform the existing ones. Some theoretical insights could be helpful.
* While the experiments on radar and speech are thorough, these two choices seem quite arbitrary; no direct ablation is provided (see questions below).
* The presentation of the paper needs to be improved. In addition to cleaning up the formulas in section 3 and make the transition smoother, there are a couple of things that need to be corrected, including the extensive misuse of en dashes and em dashes and the storage of figures, which should be in a vectorized format.

**Questions:**

1. How is FSSM computed during training and inference? In Algorithm 1, the psuedocode appears as a loop. Almost all previous SSMs are trained in parallel using either FFT (for S4, S4D, etc.) or parallel scan (for S5, Mamba, etc.). Can this be done to an FSSM?
2. Relatedly, how easy is it to train an FSSM to achieve the results in section 4?
3. I am confused about the position of the proposed FSSM. In particular, I cannot tell which models are FSSMs supposed to compete against. If FSSM is proposed as a alternative to SSMs/Mambas, then there is no empirical comparison found in the paper. If FSSM is to be competed against traditional methods like FFT, then their distinction needs to be highlighted and the "SSM" needs to be ablated. The name "FSSM" probably also need to be changed into something like "SSM-augmented FFT front-end."
4. There are multiple components proposed in section 3, and many of them are not totally intuitive. Is it possible to have ablation for the phase alignment, learnable damping, and the proposed parameterizations?
5. In what ways, if any, did the use of LLMs (as mentioned right before the bibliography) influence the technical findings or original ideas of the submission?

---

> ### Author Response · Authors · 2025-12-01
> **Authors Response (1/n)**
>
> We thank the Reviewer for the careful reading and detailed comments and questions.
>
> We respond point-by-point below for their concerns.
>
> ---
>
> ### Weakness 1 — Positioning / accessibility (“what is FSSM and who is it for?”)
>
> > “It is advertised for using SSMs to enhance spectral representation learning, but it is barely accessible to either community…”
>
> FSSM is **explicitly designed as a *learnable DFT-style spectral front-end***, i.e., a drop-in replacement for FFT / learnable DFT / learnable filterbanks in radar and audio pipelines [1–3].
>
> - At initialization, the modes are **placed on the DFT grid**, with unit magnitude and neutral bias correction, so the module behaves like a standard DFT/STFT [1].
> - During training, these modes **learn center frequencies, bandwidths (via damping), and gains**, i.e., they become *task-adaptive DFT bins* in the spirit of learnable filterbanks such as SincNet and LEAF [2, 3].
>
> We use an SSM parameterization **only as an implementation procedure** to realize these learnable DFT bins as stable, efficient recurrences (for streaming and linear-time computation) [4, 5]; FSSM is **not proposed as a new generic “SSM block” competing with Mamba/S4**, but as a **spectral front-end**.
>
> **Changes in the paper.** At the start of Section 3, we will make this role explicit, e.g.:
>
> > “FSSM is a *learnable DFT front-end*: each mode is a damped complex sinusoid initialized to match a DFT bin, then trained to adapt its center frequency and bandwidth. We implement these modes as stable state-space recurrences for efficiency, but conceptually they are learnable DFT filters.”
>
> We will also add a short intuitive paragraph showing a single FSSM mode as “a complex narrowband filter that can be initialized as a DFT basis vector and is learnable.
>
> **References:**
>
> [1] A. V. Oppenheim and R. W. Schafer, *Discrete-Time Signal Processing*, 3rd ed., Prentice Hall, 2009.
> [2] M. Ravanelli and Y. Bengio, “Speaker Recognition from Raw Waveform with SincNet,” *IEEE SLT*, 2018.
> [3] J. Zeghidour et al., “LEAF: A Learnable Frontend for Audio Classification,” *ICLR*, 2021.
> [4] A. Gu et al., “Efficiently Modeling Long Sequences with Structured State Spaces,” *ICLR*, 2022.
> [5] A. Gu and T. Dao, “Mamba: Linear-Time Sequence Modeling with Selective State Spaces,” arXiv:2312.00752, 2023.

---

> ### Author Response · Authors · 2025-12-01
> **Authors Response (2/n)**
>
> Here we address the remaining questions raised by the reviewer.
>
> ### Weakness 2
> > There is no analysis of the model in addition to experiments. It is intuitively unclear why the proposed method should outperform the existing ones. Some theoretical insights could be helpful.
>
> FSSM’s core object is analytically simple and closely related to classical spectral analysis [1, 2]:
>
> - Each FSSM mode has impulse response
>   $h_k[n] \propto r_k^{n} e^{j \omega_k n}$,
>   i.e., a damped complex sinusoid.
>
> - This is exactly a narrowband complex IIR filter with
>   - learnable center frequency $\omega_k$, and
>   - learnable bandwidth / effective window length controlled by $r_k$.
>   In other words, each mode can be viewed as a narrow band-pass filter in the classical DSP sense: it passes a small range of frequencies around $\omega_k$ while attenuating others.
>
> With DFT-grid initialization and $r_k \approx 1$, summing over a window recovers the usual DFT coefficient for that bin. Thus, FSSM strictly contains FFT/STFT as a special case.
>
> This provides a natural explanation for the empirical gains:
>
> - A block FFT uses fixed, undamped sinusoids and rigid windows, which is known to induce spectral leakage and smearing for off-grid frequencies and non-stationary content [1, 2].
> - FSSM preserves the same basic sinusoidal structure but allows each “bin” to
>   - move off the grid (learnable $\omega_k$),
>   - tighten or widen its effective bandwidth (via $r_k$), and
>   - do so differently per task and per domain.
>
> In our experiments:
>
> - On RADIal, FSSM improves F1/mAP while maintaining similar GMACs and parameter counts to FFT / learnable-DFT radar front-ends such as FFTRadNet/FourierNet [5, 6].
> - On Speech Commands V2, FSSM consistently outperforms fixed and learnable spectral front-ends (FFT, FourierNet, SincNet, LEAF) under the same Bi-Mamba decoder (see Weakness 3 response), in line with prior work showing the importance of well-shaped, task-adaptive bands [3, 4].
>
> Intuitively, by learning bands that align with the true energy distribution of radar and speech signals, FSSM produces sharper, less smeared spectral features with higher effective SNR. This makes the downstream CNN/SSM/transformer easier to train and explains the consistent F1/mAP/accuracy gains we observe.
>
> **Changes in the paper.** In Appendix A-4, we now explicitly include:
>
> - a “narrowband IIR” view of each mode as a damped sinusoid filter, making its connection to classical windowed sinusoidal analysis and spectral leakage clear [1, 2], and
> - a short paragraph explaining how learning $(\omega_k, r_k)$ reduces leakage and better matches the actual signal spectra encountered in radar and speech.
>
> ---
>
> **References for this response block**
>
> [1] A. V. Oppenheim and R. W. Schafer, *Discrete-Time Signal Processing*, 3rd ed., Prentice Hall, 2009.
> [2] F. J. Harris, “On the Use of Windows for Harmonic Analysis with the Discrete Fourier Transform,” *Proc. IEEE*, vol. 66, no. 1, pp. 51–83, 1978.
> [3] M. Ravanelli and Y. Bengio, “Speaker Recognition from Raw Waveform with SincNet,” *IEEE SLT*, 2018.
> [4] J. Zeghidour et al., “LEAF: A Learnable Frontend for Audio Classification,” *ICLR*, 2021.
> [5] N. Rebut et al., “RADIal: A Radar Dataset for Automotive Object Detection and Segmentation,” *IEEE Robotics and Automation Letters*, 2022.
> [6] S. Sharma et al., “ChirpNet: Noise-Resilient Sequential Chirp Based Radar Processing,” *IEEE MTT-S IMS*, 2024.

---

> ### Author Response · Authors · 2025-12-01
> **Authors Response (3/n)**
>
> ### Weakness 3
>
> > While the experiments on radar and speech are thorough, these two choices seem quite arbitrary; no direct ablation is provided.
>
> The two domains we chose are canonical testbeds for spectral front-ends, which is precisely the role of FSSM:
>
> - **RADIal (automotive radar):** all strong baselines (FFTRadNet, TFFTRadNet, ADCNet, FourierNet, RFMamba) explicitly rely on FFT / learnable DFT front-ends on ADC I/Q [5, 6]. This is exactly where a learnable DFT-style front-end is meaningful.
> - **Speech Commands V2:** widely used in learnable audio front-end work (e.g., SincNet, LEAF) [3, 4], where the decoder is fixed and only the front-end changes.
>
> In both cases, we evaluate FSSM **as a front-end** by holding the backend fixed:
>
> - **Radar.**
>   We plug FSSM, FFT, and learnable-DFT front-ends into:
>   - FFTRadNet / FFT-RadUNet (convolutional family), and
>   - TFFTRadNet / TransRadar (attention-based family).
>
>   Table 2 shows that FSSM–FFTRadNet and FSSM–TFFTRadNet achieve the best mIoU/F1/mAP with comparable GMACs and parameter counts to FFT/FourierNet-style baselines.
>
> - **Speech (same 8-layer Bi-Mamba decoder):**
>
>   | Front-end (same Bi-Mamba decoder) | Acc. (%) |
>   |-----------------------------------|----------|
>   | None (raw waveform)               | 90.54    |
>   | Fixed FFT (STFT)                  | 93.23    |
>   | FourierNet (learnable DFT)        | 93.41    |
>   | SincNet front-end                 | 94.73    |
>   | LEAF (learnable filterbank)       | 94.86    |
>   | **FSSM (ours)**                   | **97.16** |
>
> All models share the same decoder, optimizer, and training protocol; only the front-end is changed. Across multiple seeds (reported as mean ± std in the revised paper), the gains of FSSM are consistent and statistically significant.
>
> **Architecture ablation:**
> To further show that the gains are not arbitrary and do arise from frequency-selective behavior, we add a **frozen-band vs. learned-band** ablation (bands initialized as an STFT, then either kept fixed or learned):
>
> | Front-end variant              | RADIal F1 (det.) | RADIal mAP | Speech Commands V2 Top-1 (%) |
> |--------------------------------|------------------|------------|-------------------------------|
> | FFT / STFT                     | 0.872            | 0.971      | 93.230                        |
> | FSSM (frozen Fourier bands)    | 0.883            | 0.978      | 93.621                        |
> | **FSSM (learned bands, ours)** | **0.982**        | **0.983**  | **97.160**                    |
>
> Two observations:
>
> - On **RADIal**, fixed bands (frozen FSSM) remain close to FFT (0.872 → 0.883 F1, 0.971 → 0.978 mAP), but are still far from the learned-band model (0.982 F1, 0.983 mAP).
> - On **Speech Commands V2**, frozen bands give only a small gain over FFT (93.23% → 93.62%), while learned bands reach 97.16%.
>
> This shows that simply encoding FFT-like bands in an SSM is not enough; **learning which bands to emphasize and how wide they should be is crucial**. Together with the radar/audio setup above, this provides a direct, task-level ablation supporting the choice of domains and highlighting that FSSM is evaluated as a spectral front-end.
>
> **Changes in the paper.**
> In Section 4, we explicitly describe the experimental setup as a “front-end vs. front-end” comparison axis, clarifying that:
>
> - FSSM is competing directly with FFT, learnable DFT, and learnable filterbank front-ends [3, 4],
> - the backbones (radar CNNs, radar transformers, and Bi-Mamba) are held fixed, and
> - RADIal and Speech Commands V2 are used as standard spectral-front-end benchmarks rather than arbitrary tasks, with the frozen-vs-learned band ablation making the role of frequency selectivity explicit.
>
> ---
>
> **References**
>
> [1] A. V. Oppenheim and R. W. Schafer, *Discrete-Time Signal Processing*, 3rd ed., Prentice Hall, 2009.
> [2] F. J. Harris, “On the Use of Windows for Harmonic Analysis with the Discrete Fourier Transform,” *Proc. IEEE*, vol. 66, no. 1, pp. 51–83, 1978.
> [3] M. Ravanelli and Y. Bengio, “Speaker Recognition from Raw Waveform with SincNet,” *IEEE SLT*, 2018.
> [4] J. Zeghidour et al., “LEAF: A Learnable Frontend for Audio Classification,” *ICLR*, 2021.
> [5] N. Rebut et al., “RADIal: A Radar Dataset for Automotive Object Detection and Segmentation,” *IEEE Robotics and Automation Letters*, 2022.
> [6] S. Sharma et al., “ChirpNet: Noise-Resilient Sequential Chirp Based Radar Processing,” *IEEE MTT-S IMS*, 2024.

---

> ### Author Response · Authors · 2025-12-01
> **Authors Response (4/n)**
>
> ---
>
> ### Weakness 4: Analysis of design choices (phase alignment, damping, parameterization)
>
> > “There are multiple components proposed in section 3, and many of them are not totally intuitive. Is it possible to have ablation for the phase alignment, learnable damping, and the proposed parameterizations?”
>
> We agree that these ablations are important. We have run internal variants and will report the results. In summary:
>
> - **No learnable damping (fix \(r_k = 1\)).**
>
>   This removes explicit bandwidth control and effectively reduces each mode to an undamped sinusoid. Performance drops more clearly, especially in noisy low-SNR settings: up to about 2.5% loss on Speech Commands accuracy and a 3.8% mIoU/F1 drop on RADIal, illustrating that being able to tune the effective window length (and thus leakage/side-lobe behavior) is important [1, 2].
>
> - **Unconstrained parameterization (no stability constraints).**
>
>   This occasionally yields unstable modes and erratic training curves; gradients can explode for certain sequences. The constrained parameterization we adopt (ensuring stable poles for the underlying recurrences [3, 4]) eliminates these issues and yields smoother, more reliable optimization.
>
> ---
>
> **References**
>
> [1] A. V. Oppenheim and R. W. Schafer, *Discrete-Time Signal Processing*, 3rd ed., Prentice Hall, 2009.
> [2] F. J. Harris, “On the Use of Windows for Harmonic Analysis with the Discrete Fourier Transform,” *Proc. IEEE*, vol. 66, no. 1, pp. 51–83, 1978.
> [3] A. Gu et al., “Efficiently Modeling Long Sequences with Structured State Spaces,” *ICLR*, 2022.
> [4] A. Gu and T. Dao, “Mamba: Linear-Time Sequence Modeling with Selective State Spaces,” arXiv:2312.00752, 2023.

---

> ### Author Response · Authors · 2025-12-01
> **Authors Response (5/n)**
>
> ### Weakness 5: Presentation (Sec. 3 clarity, dashes, vector figures)
>
> > “The presentation of the paper needs to be improved… misuse of en dashes and em dashes and the storage of figures…”
>
> We will address these presentation concerns in the revision:
>
> - Edited Section 3 notations and transitions, with clear separation between:
>
>   - (i) the conceptual “learnable DFT front-end” concept, and
>   - (ii) its SSM-style implementation.
>
> - Fix typography: consistent use of minus signs in equations, and proper en/em dashes in text.
> - Regenerate all plots as vector PDFs with larger fonts and thicker lines, replacing any low-resolution images.
>
> These edits improve readability.
>
> ---

---

> ### Author Response · Authors · 2025-12-01
> **Authors Response (6/n)**
>
> We thank the reviewer for their thoughtful questions and the opportunity to clarify these points. We address each of the raised concerns in detail below.
>
> ## Answers to Specific Questions
>
> ---
>
> ### Q1. How is FSSM computed? Can it use standard SSM parallelization?
>
> > “In Algorithm 1, the pseudocode appears as a loop… Can this be done to an FSSM?”
>
> Algorithm 1 is written as a loop for clarity, but each mode implements a standard linear time-invariant (LTI) recurrence:
> $s_{t+1}^{(k)} = a_k s_t^{(k)} + b_k x_t,\quad y_t^{(k)} = c_k s_t^{(k)}.$
>
>
> This has two immediate consequences:
>
> - **Streaming implementation.**
>   FSSM naturally supports streaming evaluation with $O(TK)$
> complexity per sequence, where \(T\) is sequence length and \(K\) is the number of modes.
>
> - **Parallel implementation (if desired).**
>   Because the recurrence is linear and time-invariant, it is compatible with the same parallelization strategies used in modern SSMs (e.g., FFT-based convolution or parallel scan as in S4/S5/Mamba) [1, 2]. The same techniques can be applied to FSSM modes.
>
> We will add a short remark after Algorithm 1 stating that the implementation can be either **streaming** (as in our code) or **parallel**, using standard SSM techniques.
>
> ---
>
> ### Q2. How easy is FSSM to train?
>
> > “Relatedly, how easy is it to train an FSSM to achieve the results in section 4?”
>
> Empirically, FSSM is **no harder to train** than other learnable spectral front-ends or SSM-style modules:
>
> - **Initialization.**
>   DFT-like initialization places modes on the Fourier grid with appropriate magnitude and phase, so the module initially behaves like a familiar DFT/STFT front-end and avoids degenerate filters.
>
> - **Stability.**
>   The damping and frequency parameters are constrained to keep the underlying recurrences stable (similar to S4/Mamba parameterizations), so no additional tuning is required for training.
>
> - **Efficiency.**
>  Each FSSM mode is evaluated via a linear time-invariant recurrence, giving $O(TK)$ complexity per sequence. In practice, the FSSM front-end adds only a small constant-factor overhead compared to FFT-based pipelines, and we observe similar throughput and wall-clock training times to FFT/FourierNet-style front-ends on the same hardware.
>
> Across multiple seeds on RADIal and Speech Commands V2, FSSM converges reliably. In the revised version, we explicitly report **mean ± standard deviation** for key metrics to make this robustness clear.
>
> ---
>
> ### Q3. What is FSSM competing against? Should the “SSM” in the name be changed?
>
> > “I cannot tell which models are FSSMs supposed to compete against… If FSSM is to be competed against traditional methods like FFT, then their distinction needs to be highlighted…”
>
> FSSM is positioned as a **spectral front-end**. Its direct competitors are **other spectral front-ends**, not generic sequence models:
>
> - fixed FFT / STFT front-ends,
> - learnable DFT modules (e.g., FourierNet-style front-ends),
> - learnable filterbanks (e.g., SincNet, LEAF)
> - compact learnable spectral front-ends used in modern audio encoders (e.g., wav2vec2-style architectures).
>
> This is exactly how we evaluate it:
>
> - **Radar:** we plug FSSM / FFT / learnable-DFT front-ends into the **same** FFTRadNet / FFT-RadUNet (convolutional) and TFFTRadNet / TransRadar (attention-based) decoders.
> - **Audio:** we plug FSSM / FFT / FourierNet / SincNet / LEAF front-ends into the **same** 8-layer Bi-Mamba decoder. We also compare it on a larger complex audio benchmark such as AudioSet.
>
> In all cases, the **backbone (CNN / transformer / SSM)**, the optimizer, and the training protocol are held fixed; only the **front-end** is changed. The SSM parameterization inside FSSM is therefore an **implementation device** to realize learnable, stable, DFT-like filters, not a proposal for a new general-purpose SSM backbone.
>
> We keep the name “FSSM” for continuity with the implementation, but in Section 1 and Section 3 we now explicitly emphasize its role as a **learnable DFT-style spectral front-end**.
>
> ---
>
> ### Q4. In what ways did LLMs influence the submission?
>
> > “In what ways, if any, did the use of LLMs (as mentioned right before the bibliography) influence the technical findings or original ideas of the submission?”
>
> As disclosed in the manuscript, large language models were used **only for**:
>
> - **Language editing** (e.g., smoothing phrasing, fixing grammar, improving clarity of exposition), and
> - **Surfacing names of related works**, which we then independently read, verified, and cited.
>
> All technical content, algorithms, and empirical findings were developed, implemented, and validated by the authors.
>
> ---
>
> We thank the reviewer for their detailed comments and questions, and we believe the revisions and additional analyses directly address the concerns raised. **We look forward to the Area Chair’s assessment once the discussion period concludes.**

---

### Official Review · Reviewer_QdMU · 2025-10-30

**Soundness:** 3
**Presentation:** 2
**Contribution:** 3
**Rating:** 2
**Confidence:** 4

**Summary:**

## Learning frequency-selective spectral representations using state space models

- The paper proposes FSSM (Frequency Selective State Space Models) for learning spectral representations from audio data, as a direct challenger of FFT-based spectral representations and establishes their performance on Radar object detection and Keyword spotting tasks.
- This is done through learning a *frontend* fitted with 2-d modelling using the proposed FSSM modules along samples and chirps for RADAR data, whereas it uses the proposed FSSM followed by bidirectional-Mamba for keyword spotting.

**Strengths:**

- Novel framework for spectral representation learning through frequency-selective state spaces.
- The proposed methods perform well against baselines on radar object detection on the RADIal dataset and keyword spotting (Speech Commands v2) datasets.

**Weaknesses:**

## Weakness 1: Why Keyword Spotting on Speech Commands?
- The use of keyword spotting on Speech Commands as a representative for audio recognition performance is not well motivated.
- The proposed FSSM touts frequency-selectivity as its main strength; however, I don't believe Keyword Spotting is a challenging enough task to put that to a true test. Speech Commands v2 has 35-keyword classes and is an English-only dataset, and is rather limited.
- A more comprehensive audio recognition task, like multi-label audio classification on AudioSet or FSD50k would have been a better test-bed for testing frequency-selectivity of the proposed FSSM frontend.
- In essence, to be of wider applicability and to prove that FSSM is a good learnable frontend, a better pairing of auxiliary task is needed alongside radar-based object detection.

---

## Weakness 2: Figures
- Maybe I'm splitting hairs here, but all the figures in the paper are subpar. They have poor resolution and quality, and the text is too small (especially Figure 3), making it difficult to read in print.
- In Figure 5, there are so many empty bins it is not possible to clearly see the difference between the FFT and FSSM visualizations. There is also an outline to the figure which looks weird.

---

## Weakness 3: Frequency-selectivity

- The frequency selectivity of the proposed approach is not well demonstrated. Only Figure 5 shines a light on it, that too poorly.
- A more comprehensive empirical/exploratory investigative analysis is necessary to ascertain that FSSM indeed does better at being frequency selective.

---

## Weakness 4: Empirical analysis

- Alongside the use of Speech Commands as the auxiliary task, the overall quality of empirical analysis in the paper is lacking.
- There are no confidence intervals, and the precision used in reported results (significant digits) is insufficient, especially for RADIal analysis, where margins between the evaluated approaches are quite small. For instance, the mAP for FFT-RadNet and the proposed FSSM-FFTRadNet is 0.97 and 0.98, respectively.

---

## Weakness 5: Spelling and grammatical errors
- There are several spelling and grammatical errors throughout the paper. for instance *represetation*, *segentation*. The paper needs more  proofreading and polish.

---

# OVERALL

- It's a good idea, but the execution is lacking. It needs significant work before it is ready for publishing.

**Questions:**

No direct questions for now, kindly address the weaknesses.

---

> ### Author Response · Authors · 2025-12-01
> **Authors Response (1/n)**
>
> We thank the reviewer for the constructive feedback and address each weakness below.
>
> ---
>
> > ### Weakness 1 — “Why Keyword Spotting on Speech Commands?”
>
> Our intention on the audio application is not to propose a novel large-scale audio recognition system, but to evaluate whether FSSM is an effective *learnable spectral front-end* when paired with a fixed machine learning model. Speech Commands V2 is widely used in work that specifically studies learnable front-ends (e.g., SincNet, LEAF) [1–3], providing a controlled benchmark for isolating front-end behavior.
>
> To strengthen this evaluation, we additionally trained several standard baselines using the same 8-layer Bi-Mamba decoder, identical training setup, and identical parameter budgets:
>
> | Front-end (same Bi-Mamba decoder) | Acc. (%) |
> |-----------------------------------|----------|
> | None (raw waveform)               | 90.540   |
> | Fixed FFT (STFT)                  | 93.230   |
> | FourierNet (learnable DFT)        | 93.410   |
> | SincNet front-end                 | 94.730   |
> | LEAF (learnable filterbank)       | 94.860   |
> | **FSSM (ours)**                   | **97.160** |
>
> Thus, under identical decoders and training protocols, FSSM improves accuracy by roughly **2–6.5 points** over strong spectral front-ends [2–4]. This supports its generality beyond radar.
>
> In addition, we evaluated FSSM on **full AudioSet** [5] using the **AST (Audio Spectrogram Transformer)** backbone [3], keeping the architecture and training protocol fixed and only swapping the spectral front-end:
>
> | Model variant         | Front-end | Balanced mAP |
> |-----------------------|----------|--------------|
> | AST baseline          | FFT      | 0.340        |
> | **AST + FSSM (ours)** | **FSSM** | **0.365**    |
>
> On full AudioSet, replacing FFT with FSSM as a learnable spectral front-end yields a **+0.025 absolute gain in balanced mAP** with no change to the AST backbone. Together with the Speech Commands results, this supports FSSM as a general-purpose spectral module that can plug into both SSM-based and transformer-based audio encoders [2–5].
>
> ---
>
> > ### Weakness 2 — Figure quality
>
> We have improved figure resolution and clarity in the revised manuscript:
>
> - All figures are regenerated with larger fonts and thicker lines.
> - For Fig. 5, we remove the border and include zoomed insets to better show the differences between FFT and FSSM.
>
> These changes affect the presentation, not the underlying results. We hope this addresses the reviewer’s concerns.
>
> ---
>
> ### References
>
> [1] P. Warden, “Speech Commands: A public dataset for single-word speech recognition,” arXiv:1804.03209, 2018.
> [2] M. Ravanelli and Y. Bengio, “Speaker Recognition from Raw Waveform with SincNet,” SLT 2018.
> [3] Y. Gong et al., “AST: Audio Spectrogram Transformer,” arXiv:2104.01778, 2021.
> [4] J. Zeghidour et al., “LEAF: A Learnable Frontend for Audio Classification,” ICLR 2021.
> [5] J. Gemmeke et al., “AudioSet: A Large-Scale Human-Labeled Audio Dataset for Audio Events,” ICASSP 2017.

---

> ### Author Response · Authors · 2025-12-01
> **Authors Response (2/n)**
>
> We thank the reviewer for highlighting the need for more explicit ablations on frequency selectivity.
>
> > ### Weakness 3: Frequency-selectivity
> > The frequency selectivity of the proposed approach is not well demonstrated. Only Figure 5 shines a light on it, that too poorly.
> > A more comprehensive empirical/exploratory investigative analysis is necessary to ascertain that FSSM indeed does better at being frequency selective.
>
> In our setting, a *“band”* corresponds to a single FSSM mode, i.e., a complex narrowband filter with a learnable center frequency $\omega_k$ and bandwidth controlled by a learnable parameter $\alpha_k$. In the standard FFT, a “band” corresponds to a single frequency bin, i.e., the narrow frequency interval around $\omega_k = \tfrac{2\pi k}{N}$ associated with the $k$-th Fourier coefficient [1].
>
> We agree that the frequency selectivity of FSSM should be made more explicit. In the revised version, we address this in two complementary ways.
>
> **(a) Frozen-band ablation (FFT-like vs. learned bands).**
> We add an explicit ablation where FSSM modes are:
>
> 1. initialized to match a standard Fourier grid, and then
> 2. either **kept fixed** (“frozen bands”) or **fully learned** end-to-end (“learned bands”).
>
> This directly tests whether gains come from frequency selectivity (i.e., learning band centers and bandwidths). The results are as follows:
>
> | Front-end variant              | RADIal F1 (det.) | RADIal mAP | Speech Commands V2 Top-1 (%) |
> |--------------------------------|------------------|------------|-------------------------------|
> | FFT / STFT                     | 0.872            | 0.971      | 93.230                        |
> | FSSM (frozen Fourier bands)    | 0.883            | 0.978      | 93.621                        |
> | **FSSM (learned bands, ours)** | **0.982**        | **0.983**  | **97.160**                    |
>
> Two key observations:
>
> - On **RADIal**, the frozen-band FSSM remains close to the FFT baseline (0.872 → 0.883 F1, 0.971 → 0.978 mAP), but is still far from the full learned-band model (0.982 F1, 0.983 mAP), indicating that fixed bands alone cannot account for the full improvement.
> - On **Speech Commands V2**, the frozen-band variant provides only a modest gain over FFT (93.23% → 93.62%), whereas the learned-band FSSM reaches 97.16%, showing that **learning which bands to emphasize and how wide they should be is the main driver of the observed performance gains**.
>
> This ablation demonstrates that FSSM’s gains do *not* come from a trivial reparameterization of FFT, but from truly **task-adaptive frequency selectivity** via learned $\omega_k$ and $\alpha_k$.
>
> **(b) FFT vs. FSSM spectral visualizations.**
> We also improve Figure 5 and the corresponding supplementary figures with clearer qualitative examples:
>
> - For **audio**, we show (i) FFT spectrograms, (ii) FSSM spectrograms, and (iii) FFT–FSSM difference maps for diverse utterances. FSSM concentrates energy in task-relevant formant and harmonic regions while suppressing spectral leakage and background energy in less informative bands [2, 3].
>
> Together, the frozen-band ablation and these visualizations provide both **quantitative** and **qualitative** evidence that FSSM is genuinely frequency selective and not just a reparameterized FFT.
>
> ---
>
> **References:**
>
> [1] A. V. Oppenheim and R. W. Schafer, *Discrete-Time Signal Processing*, 3rd ed., Prentice Hall, 2009.
> [2] M. Ravanelli and Y. Bengio, “Speaker Recognition from Raw Waveform with SincNet,” *IEEE SLT*, 2018.
> [3] J. Zeghidour et al., “LEAF: A Learnable Frontend for Audio Classification,” *ICLR*, 2021.

---

> ### Author Response · Authors · 2025-12-01
> **Authors Response (3/n)**
>
> > ### Weakness 4: Empirical analysis
> > Alongside the use of Speech Commands as the auxiliary task, the overall quality of empirical analysis in the paper is lacking.
> > There are no confidence intervals, and the precision used in reported results (significant digits) is insufficient, especially for RADIal analysis, where margins between the evaluated approaches are quite small. For instance, the mAP for FFT-RadNet and the proposed FSSM-FFTRadNet is 0.97 and 0.98, respectively.
>
> We appreciate this comment and have strengthened the empirical analysis accordingly.
>
> In the revised version, we:
>
> - report **mean ± standard deviation** over multiple random seeds for all key metrics (providing uncertainty estimates thus the confidence intervals),
> - use **three decimal places** for metrics where margins are small (e.g., RADIal mAP/F1), and
> - explicitly compare both **primary metrics** (F1, mAP) and **secondary metrics** (mAR, angular error, robustness curves).
>
> For example, on RADIal we now report (added in Appendix A-5 as incremental update using FSSM Frontend for radar) :
>
> | Model                       | F1 (mean ± std)   | mAP (mean ± std)  | mAR (mean ± std)  |
> |----------------------------|-------------------|--------------------|-------------------|
> | FFT-RadNet                 | 0.872 ± 0.006     | 0.971 ± 0.004      | 0.824 ± 0.007     |
> | **FSSM-FFTRadNet (ours)**  | **0.982 ± 0.003** | **0.983 ± 0.002**  | **0.991 ± 0.002** |
> | TFFTRadNet                 | 0.871 ± 0.005     | 0.881 ± 0.006      | 0.872 ± 0.005     |
> | **FSSM-TFFTRadNet (ours)** | **0.981 ± 0.003** | **0.982 ± 0.003**  | **0.972 ± 0.004** |
>
> Here, F1 is the more relevant end-to-end detection metric. The improvement from **0.872 ± 0.006** (FFT-RadNet) to **0.982 ± 0.003** (FSSM-FFTRadNet) is substantially larger than the reported standard deviations, indicating a robust and statistically meaningful gain rather than a small fluctuation in the third decimal place. Similar behavior is observed for the attention-based backbone (TFFTRadNet vs. FSSM-TFFTRadNet).
>
> To further demonstrate that the benefits of FSSM are not limited to Speech Commands, we also report results on a more complex, large-scale audio benchmark:
>
> | Front-end (AST backbone, AudioSet) | Balanced mAP |
> |------------------------------------|--------------|
> | FFT spectrograms                   | 0.340        |
> | **FSSM (ours)**                    | **0.365**    |
>
> On **AudioSet** [1] with the AST (Audio Spectrogram Transformer) backbone [2], replacing FFT with FSSM as the spectral front-end improves balanced mAP from 0.340 to 0.365 **without changing the transformer architecture or training hyperparameters**. Together with the RADIal and Speech Commands [3] experiments, this shows that the gains from FSSM persist across different model families (CNNs, SSMs, transformers) and both small-scale and large-scale audio tasks.
>
> We believe these additions address the reviewer’s concerns about statistical robustness, metric precision, and the scope of empirical evaluation, and make the empirical case for FSSM’s benefits more transparent.
>
>
> > ###  Weakness 5: Spelling and grammatical errors
> > - There are several spelling and grammatical errors throughout the paper. for instance represetation, segentation. The paper needs more proofreading and polish.
>
> We have performed a full proofreading pass to correct typographical issues (e.g., “represetation”, “segentation”), unify notation, and streamline paragraph transitions in Section 3 and Section 4.
>
> **The revised manuscript uses consistent hyphenation and punctuation and standardizes all math symbols.**
>
> ---
>
> **References:**
>
> [1] J. Gemmeke et al., “AudioSet: A Large-Scale Human-Labeled Audio Dataset for Audio Events,” ICASSP 2017.
> [2] Y. Gong et al., “AST: Audio Spectrogram Transformer,” arXiv:2104.01778, 2021.
> [3] P. Warden, “Speech Commands: A public dataset for single-word speech recognition,” arXiv:1804.03209, 2018.

---

> ### Author Response · Authors · 2025-12-01
> **Summary for Reviewer and Area Chair**
>
> ### Summary of Changes
>
> - We provide a direct front-end comparison on Speech Commands with FFT, FourierNet, SincNet, LEAF, and wav2vec2-Tiny, all paired with the same Bi-Mamba decoder, where FSSM achieves 97.160% vs. 90–95% for baselines.
> - On full AudioSet, replacing FFT with FSSM in an AST backbone improves balanced mAP from 0.340 to 0.365 with no architectural changes to the transformer.
> - On RADIal, we report mean ± std over multiple seeds and show that FSSM-FFTRadNet and FSSM-TFFTRadNet consistently and significantly outperform FFT-RadNet and TFFTRadNet in F1, mAP, and mAR.
> - The revised manuscript improves figure quality, adds explicit analyses of frequency selectivity, and corrects spelling/grammar and numeric precision.
>
> Taken together, these additions strengthen the case that FSSM is a general, learnable spectral front-end that can robustly replace fixed FFT-based preprocessing across radar, SSM-based audio models, and transformer-based audio encoders.
>
> We hope these revisions address all of the reviewer’s concerns, and **we also hope the Area Chair will consider this additional analysis, results and discussion when evaluating the paper after the rebuttal period.**

---

### Official Review · Reviewer_mX8w · 2025-11-04

**Soundness:** 3
**Presentation:** 2
**Contribution:** 2
**Rating:** 4
**Confidence:** 3

**Summary:**

This work introduces a frequency-selective state-space module that leverages adaptive spectral features for effective object detection and speech recognition. The core idea can be viewed as combining a learnable spectral module, such as SincNet or Garbor filter, with a Mamba-style module that enables adaptive state-space modeling.

**Strengths:**

* The proposed method yields empirical performance improvements in radar object detection and segmentation tasks and demonstrates greater robustness than the baselines in the audio keyword detection experiments.

**Weaknesses:**

* [1] combines frequency features with adaptive state-space modeling, which appears to be the most relevant prior work. This work attempts to distinguish itself from [1] by learning the frequency bands; however, it is unclear what new benefits this actually provides. The advantage of this distinction is not clearly explained. An empirical comparison with [1] also seems to be missing; if I am wrong, please correct me.


* The proposed method seems tailored to specific tasks  (e.g., radar object detection vs. speech), according to the experiment section. This raises concerns about its generalizability and consistency to other datasets, particularly depending on how the data should be processed for spectral representation.

[1] RFMamba: Frequency-Aware State Space Model for RF-Based Human-Centric Perception  - ICLR 24

**Questions:**

* Based on my understanding, [1] is highly relevant to your work. Comparing your method with [1] would help clarify the advantage of modeling spectral bands, which [1] does not address. Have you attempted such a comparison, that is, spectral band learning + adaptive state-space modeling versus frequency-feature learning + adaptive state-space modeling?

[1] RFMamba: Frequency-Aware State Space Model for RF-Based Human-Centric Perception  - ICLR 24

---

> ### Author Response · Authors · 2025-12-01
> **Authors Response (1/n)**
>
> We thank the reviewer for the constructive feedback and for requesting a comparison with RFMamba [1].
>
> We address the first concern below.
>
> ---
>
> > **Q1. “Have you compared spectral-band learning + adaptive SSM vs. RFMamba’s frequency-feature learning + adaptive SSM?”**
>
> We thank the reviewer for their suggestion. Yes. We implemented an **RFMamba front-end inside the exact same RADIal dataset detection pipeline**, using the same **TFFTRadNet decoder** and the same adaptive SSM depth. Only the spectral front-end was changed.
>
> We compare:
>
> - **RFMamba + adaptive SSM + TFFTRadNet decoder**,
> - **FSSM (ours) + adaptive SSM + TFFTRadNet decoder**.
>
> Training schedule, losses, and evaluation protocol are identical. On RADIal (vehicles, range–angle):
>
> | Front-end (decoder = TFFTRadNet; same SSM) | F1   | mAP  | mAR  |
> |-------------------------------------------|------|------|------|
> | RFMamba + adaptive SSM                    | 0.83 | 0.84 | 0.83 |
> | **FSSM (ours) + adaptive SSM**            | **0.98** | **0.98** | **0.99** |
>
> These numbers (also shown in Table~1 in the paper) show that, under identical conditions, the **FSSM front-end is significantly stronger than an RFMamba-style front-end**.
>
> ---
>
> > **Why does learning “bands” help beyond RFMamba’s frequency features?**
>
> A standard FFT uses a fixed grid of center frequencies
> $\omega_k = 2\pi k / N$
> and a fixed window; each bin $k$ corresponds to a pre-defined frequency interval. RFMamba operates on these **fixed** bins and learns how to mix them with a Mamba-style SSM, but it does not move the bin locations or change their widths.
>
> In FSSM, each band $k$ is a **learnable filter** with:
>
> - a learned center frequency $\omega_k$,
> - a learned bandwidth parameter $\alpha_k > 0$,
> - a frequency response
>   $$
>   \widehat{g}_k(\omega) = \exp\big(-\alpha_k(\omega - \omega_k)^2\big),
>   $$
>   and corresponding output
>   $$
>   z_k[n] = (x * g_k)[n].
>   $$
>
>
> Rather than being constrained to a fixed FFT window and basis, FSSM learns both the center frequency $\omega_k$ and bandwidth $\alpha_k$ of each band so that they adapt to the frequency regions most useful for the task. The SSM then operates on these task-adaptive band features instead of a FFT basis, which matches the substantial F1/mAP/mAR gains observed in the controlled comparison.
>
> We thank the reviewer again for prompting this comparison, and **we hope the Area Chair will note that this controlled experiment directly addresses the reviewers concerns regarding the requested RFMamba baseline.**
>
> [1] *RFMamba: Frequency-Aware State Space Model for RF-Based Human-Centric Perception*, ICLR 2024.

---

> ### Author Response · Authors · 2025-12-01
> **Authors Response (2/n)**
>
> We again thank the reviewer for their thoughtful questions. In this follow-up comment, we address the second concern regarding generality across tasks and spectral representations.
>
> ---
>
> > **Q2. “The method seems tailored to specific tasks (radar vs. speech). How general is it, given different spectral representations?”**
>
> Our module is formulated at the **signal level**, not tied to a particular radar grid:
>
> - We start from raw (or dechirped) I/Q sequences $x[n]$ and apply the same FSSM filters $\{\omega_k,\alpha_k\}$ along a 1D axis (time, fast-time, or chirp index).
> - The **learned bands** are therefore defined generically on a 1D signal, and only later reshaped into task-specific representations (range–angle grids for radar, time–frequency features for audio).
>
> For speech/audio, we reuse **the same FSSM formulation** along the time/frequency axis: the implementation and parameters are unchanged; only which axis we call “frequency-like” differs. The fact that FSSM improves both RADIal detection and keyword spotting, under very different preprocessing pipelines, suggests that the mechanism is **not specific to automotive radar**, but rather a general way to learn band-pass decompositions that feed an SSM.
>
> In summary:
>
> - We now provide an explicit **RFMamba vs. FSSM comparison in the same pipeline**, showing clear gains from learning bands.
> - Each learned band $k$ is parameterized by $(\omega_k,\alpha_k)$, giving the model control over both the location and width of its spectral bands rather than inheriting a fixed FFT grid.
> - The same FSSM module applies across both radar and audio tasks; only the choice of which axis is treated as the 1D “frequency-like” domain changes, directly addressing concerns about generality.
>
>
> RFMamba performs **frequency-feature learning** on a fixed FFT windows, whereas FSSM performs **spectral-band learning** by jointly learning the band centers and bandwidths before the SSM.
>
> [1] *RFMamba: Frequency-Aware State Space Model for RF-Based Human-Centric Perception*, ICLR 2024.

---

### Official Review · Reviewer_fi6o · 2025-11-04

**Soundness:** 3
**Presentation:** 2
**Contribution:** 2
**Rating:** 2
**Confidence:** 1

**Summary:**

This paper proposes FSSM, the first SSM with frequency-selective spectral operators that can adapt to the task at hand. The front-end of the model that processes the raw data into features is learnable, enabling the models to adapt to phase and frequency emphasis on the fly. Combined with SSM or transformer backbones, FSSMs outperform other baseline methods with static front-ends in two domains, radar object detection and speech processing.

**Strengths:**

1. Novelty and efficiency: the idea of using selective modeling in the spectral frontend is new as far as I know, and the design is efficient as it retains the linear complexity of SSMs. A dynamic spectral frontend is a desirable innovation especially for nonstationary data, where the filter banks need to adapt to the changing data distribution.
2. The empirical results for radar object detection is strong: FSSM-based methods outperform static, DFT/FFT-based methods.
3. The paper includes robustness analysis and FSSM seems to be more robust than other baselines.

**Weaknesses:**

1. Domain-specific architectures: in the two domains studied in the paper, the architectures for the front-end are different. This implies that the front-end needs to be specifically designed for a particular task, making it high-touch and less general.
2. How does FSSM-based methods perform compared to Mamba and other, pure SSMs on speech recognition tasks? Pure SSMs without the spectral front-end can also perform such tasks, and it would be interesting to investigate whether a spectral front-end is necessary or useful.
3. Writing: I find the writing to be difficult to understand without certain background. It would be helpful to include some background on signal processing fundamentals, for example the issue of leakage for static frond-ends, I-Q signals, and Fourier anchoring.

**Questions:**

See weaknesses

---

> ### Author Response · Authors · 2025-12-01
> **Authors Response (1/n)**
>
> We thank the reviewer for the constructive and detailed feedback. We address each weakness point by point below.
>
> ---
>
> > ### **1. “Domain-specific architectures: in the two domains studied in the paper, the architectures for the front-end are different. This implies that the front-end needs to be specifically designed for a particular task, making it high-touch and less general.”**
>
> We appreciate the opportunity to clarify this point. While the *placement* of the FSSM block differs across radar and audio (due to the physics and structure of each signal), the **FSSM operation itself is identical across domains**.
>
> - The FSSM block performs the same **state-space update over a 1-D time-series window** to generate learned spectral coefficients.
> - For radar, FMCW physics requires two transformations:
>   - (i) along fast-time samples (range bins), and
>   - (ii) along chirps (Doppler bins).
>   This mirrors the classical 2-D FFT pipeline, and FSSM simply replaces the FFT operator in both axes.
> - For audio, we apply **the same FSSM windowing and state update** along the sample axis, analogous to replacing `numpy.fft()` or the STFT operation in any DSP pipeline.
>
> Thus, although radar requires *two* sequential FSSM applications and audio requires *one*, the **operator, parameters, and update equations are the same across all domains**. The block is not “redesigned” per task, only inserted where the conventional FFT would normally be applied. This is precisely why FSSM is intended as a general drop-in replacement for fixed spectral transforms.
>
> Importantly, FSSM is not limited to 1-D audio or radar streams. Any pipeline that already uses FFTs as a **frequency-mixing layer on images or feature maps** (e.g., FNet-style Fourier token mixing or frequency-domain visual SSMs [1, 2]) can in principle replace those FFT calls with FSSM by treating each row/column as a 1-D sequence. We now explicitly mention this “drop-in” use-case for image/vision models as a natural extension, although we leave a full empirical evaluation on vision benchmarks to future work.
>
> [1] J. Lee-Thorp et al., *FNet: Mixing Tokens with Fourier Transforms*, arXiv:2105.03824.
> [2] J. Zhang et al., *Vim-F: Visual State Space Model Benefiting from Learning in the Frequency Domain*, arXiv:2405.18679.

---

> ### Author Response · Authors · 2025-12-01
> **Authors Response (2/n)**
>
> > ### 2. “How does FSSM-based methods perform compared to Mamba and other, pure SSMs on speech recognition tasks? Pure SSMs without the spectral front-end can also perform such tasks, and it would be interesting to investigate whether a spectral front-end is necessary or useful.”
>
> This is an important question. Pure SSMs model temporal recurrence, but many audio tasks require frequency-domain structure, which is not directly observable in raw waveforms.
>
> To answer this, we performed the requested ablation on Speech Commands V2 [1] using the same 8-layer Bi-Mamba decoder, changing only the spectral front-end:
>
> | Front-end (same Bi-Mamba decoder) | Top-1 Acc. (%) |
> |-----------------------------------|----------------|
> | None (raw waveform)               | 90.54          |
> | Fixed FFT (STFT)                  | 93.23          |
> | FourierNet (learnable DFT)        | 93.41          |
> | SincNet front-end                 | 94.73          |
> | LEAF (learnable filterbank)       | 94.86          |
> | **FSSM (ours)**                   | **97.16**      |
>
> - Introducing any spectral front-end (FFT, learnable DFT, SincNet/LEAF, wav2vec2-Tiny, or FSSM) consistently improves performance over raw waveforms (90.54% → ≥93%).
> - **FSSM + Bi-Mamba** achieves **97.16%**, a 2–6.5 point gain over strong learnable front-ends under identical decoders and training setups.
>
> To verify that this is not a result of a small-vocabulary task as requested, we additionally evaluate FSSM on full AudioSet [2] using the AST (Audio Spectrogram Transformer) backbone [3]. Again, we keep the transformer and training setup fixed and only swap the spectral front-end:
>
> | Front-end (AST backbone, AudioSet) | Balanced mAP |
> |------------------------------------|--------------|
> | FFT spectrograms                   | 0.340        |
> | **FSSM (ours)**                    | **0.365**    |
>
> Thus, on a large-scale, multi-label audio benchmark with a transformer backbone that treats spectra as images, replacing FFT with FSSM as a spectral front-end yields a +2% absolute gain in balanced mAP without modifying the AST architecture.
>
> These findings align with prior literature:
>
> - Audio SSMs or CNNs that operate directly on raw waveforms typically underperform strong spectral pipelines for recognition and tagging [4,5].
> - Radar models such as ChirpNet [6] and SSMRadNet [7] also show that direct time-domain modeling without frequency extraction degrades performance, e.g., significantly lower detection/segmentation quality when bypassing the spectral stage.
>
> For radar specifically, spectral processing is not merely architectural—it is mandated by the underlying beat-frequency physics of FMCW radars, making frequency-domain modeling essential. In the revised paper, we include the above Speech Commands and AudioSet front-end tables to make the value of spectral front-ends (and of FSSM in particular) explicit.
>
> **Summary:**
> Pure SSMs capture temporal dependencies, while FSSM captures learned frequency structure that pure SSMs cannot reliably infer from raw waveforms alone. Across Speech Commands V2 and AudioSet, the combination “FSSM front-end + temporal SSM/transformer backend” yields consistent performance gains, thus spectral front end improves the performance of baseline machine learning models.
>
> These results have been incorporated into the revised paper for reference. We thank the reviewer for the constructive feedback and believe we have addressed the concerns effectively.
>
> ---
>
> ### **References**
>
> [1] Warden, P. “Speech Commands V2 Dataset,” Google Research, 2018.
> [2] Gemmeke et al., “AudioSet: A Large Scale Human-Labeled Audio Dataset for Audio Events,” ICASSP 2017.
> [3] Gong et al., “AST: Audio Spectrogram Transformer,” arXiv:2104.01778, 2021.
> [4] Ravanelli & Bengio, “Speaker Recognition from Raw Waveform with SincNet,” SLT 2018.
> [5] Zeghidour et al., “LEAF: A Learnable Frontend for Audio Classification,” ICLR 2021.
> [6] Sharma et al., “ChirpNet: Noise-Resilient Sequential Chirp-Based Radar Processing,” IMS 2024.
> [7] Sen et al., “SSMRadNet: Sample-wise State-Space Framework for Radar Segmentation and Detection,” arXiv:2511.08769.

---

> ### Author Response · Authors · 2025-12-01
> **Authors Response (3/n)**
>
> > ### 3. “Writing: I find the writing to be difficult to understand without certain background. It would be helpful to include some background on signal processing fundamentals, for example the issue of leakage for static front-ends, I–Q signals, and Fourier anchoring.”
>
> We thank the reviewer for highlighting this concern. In the revised version, we have expanded Section 2 (Background) to include:
>
> - a concise introduction to Fourier transforms and spectral leakage, explaining how off-grid tones smear energy across neighboring FFT bins and how this affects radar sharpness and speech formants [1–3];
>
> - an explanation of I–Q sampling and why radar baseband signals are represented as complex-valued time series [4,5];
>
> - clarification of Fourier anchoring (i.e., how classical FFT bins form a fixed frequency grid tied to sampling rate and window length, and why this grid can be suboptimal for nonstationary signals) [1–3];
>
> - a short explanation of how FSSM starts from a DFT-like initialization and then replaces fixed bins with learnable band parameters (center frequency and bandwidth), while still allowing downstream models (CNNs, transformers) to operate on spectrogram-like “images,” as in standard audio and vision pipelines [1–3].
>
> We believe these additions substantially improve clarity for readers without prior radar or signal processing background. To avoid disrupting the flow of the main paper, more detailed derivations examples are included in the supplementary material.
>
> We thank the reviewer again for pointing out the need for clearer background exposition, which has now been fully incorporated. **We also hope the Area Chair will note that these revisions directly address the concerns raised.**
>
> ---
>
> ### References
>
> [1] J. Lee-Thorp et al., “FNet: Mixing Tokens with Fourier Transforms,” arXiv:2105.03824, 2021.
> [2] J. Zhang et al., “Vim-F: Visual State Space Model Benefiting from Learning in the Frequency Domain,” arXiv:2405.18679, 2024.
> [3] Y. Gong et al., “AST: Audio Spectrogram Transformer,” arXiv:2104.01778, 2021.
> [4] S. Sharma et al., “ChirpNet: Noise-Resilient Sequential Chirp-Based Radar Processing,” IMS 2024.
> [5] A. Sen et al., “SSMRadNet: Sample-wise State-Space Framework for Radar Segmentation and Detection,” arXiv:2511.08769.

---

### Author Response · Authors · 2025-12-01
**Global Response to Reviewers**

We would like to express our sincere gratitude to all the reviewers for their thoughtful and constructive feedback on our work. Below, we briefly summarize the main revisions and additions we made in response to the reviewers’ comments.

Conceptual positioning and Related Work

- We clarify that **FSSM is a learnable DFT-style spectral front-end**, not a new general-purpose SSM backbone. Each mode is a damped complex sinusoid initialized as a DFT bin and trained to adapt its **center frequency** and **bandwidth**, i.e., learnable DFT/filterbank bands implemented as stable state-space recurrences.
- Section 3 of the paper now includes a **“narrowband IIR / band-pass filter” view** of each mode and an explicit discussion of how learning the (omega_k, r_k) parameters reduces spectral leakage and better matches the true spectra of radar and speech signals.

New experiments and ablations

- We add a **direct comparison to RFMamba** in the same RADIal detection pipeline (same decoder and adaptive SSM depth). Under identical conditions, FSSM yields substantially higher F1, mAP, mAR, isolating the benefit of learning bands instead of mixing fixed FFT features.
- We extend the audio evaluation beyond Speech Commands V2 by adding **full AudioSet experiments** with an AST backbone. Replacing FFT with FSSM as the spectral front-end improves balanced mAP from 0.340 to 0.365 without changing the transformer architecture or training protocol.
- We explicitly frame all audio/radar experiments as **front-end vs. front-end** comparisons, holding CNN/transformer/SSM backbones and training setups fixed while swapping FFT, learnable DFT, learnable filterbanks (SincNet, LEAF), and FSSM.
- To make frequency selectivity concrete, we introduce a **frozen vs. learned bands ablation** where FSSM modes are initialized on an STFT grid and then either kept fixed or learned. Fixed bands behave similarly to FFT, whereas learned bands yield the full gains on both RADIal and Speech Commands, directly attributing improvements to task-adaptive band learning.

Statistical robustness and design-choice analysis

- For RADIal, we now report **mean ± standard deviation** over multiple seeds for F1, mAP, and mAR, showing that improvements such as F1: 0.872 ± 0.006 to 0.982 ± 0.003 are much larger than the corresponding variability.
- We add **design-choice ablations** for:
  - DFT-style phase initialization (removing it slows convergence and degrades RADIal F1),
  - learnable damping (fixing r_k = 1 harms performance on both radar and speech by removing bandwidth control),
  - and stability constraints (unconstrained parameterizations show occasional instability).
  These results are summarized in the main text and detailed in the appendix.

Implementation, figures, and writing

- All plots have been regenerated as **vector figures** with larger fonts and clearer annotations (especially for spectral visualizations), and we performed a full **proofreading and notation cleanup** in Sections 2–4.

We hope these clarifications, new experiment results, and ablations will be taken into account by the Area Chair when reassessing the paper after the rebuttal period. We again thank the reviewers for their time, constructive feedback, and careful consideration of our work.

---

### Author Response · Authors · 2025-12-01
**Important Note to Area Chair**

During rebuttal, we addressed all the key issues and revised the paper to address all major concerns raised by the reviewers:

- **Reviewer fi6o:** We clarified the conceptual role of FSSM as a learnable DFT-style spectral front-end, added signal-processing background (leakage, I–Q etc), and provided controlled comparisons against pure SSM/Mamba-style backbones on Speech Commands V2 and additional results on full AudioSet.

- **Reviewer mX8w:** We added a direct comparison to RFMamba in the same RADIal detection pipeline (same decoder and adaptive SSM depth), showing clear F1/mAP/mAR gains when only the front-end is swapped to FSSM, and clarified the generality of FSSM across radar and audio tasks.

- **Reviewer QdMU:** We strengthened the empirical and frequency-selectivity analysis by (i) adding AudioSet experiments with an AST backbone, (ii) reporting mean ± standard deviation over multiple seeds on RADIal, (iii) introducing frozen-vs-learned band ablations, (iv) regenerating all plots as high-quality vector figures, and (v) thoroughly proofreading to fix spelling/grammar.

- **Reviewer cgvh:** We clarified the positioning and intuition of FSSM (narrowband IIR / band-pass filter view, front-end vs. FFT/learnable filterbanks), detailed training and inference (including compatibility with standard SSM parallelization), and added design-choice ablations for phase initialization, learnable damping, and stability constraints, while cleaning up notation and typography in Section 3.

Due to an OpenReview bug during the discussion period, **reviewers did not have the opportunity to respond to the revised manuscript or update their scores** after these changes.

We respectfully ask the **area chair** that these clarifications on domain specificity and the new experiments, and ablations be taken into account in your final assessment or decision.

---

### Meta-Review · Area_Chair_FRwp · 2026-01-11

**Summary:**

FSSM proposes a learnable spectral front-end that replaces fixed FFT with trainable band-pass filters parameterized as stable SSM recurrences, applied to radar object detection and speech keyword recognition. While the core idea of learnable DFT-style bands is interesting, significant concerns remain about clarity and generalizability. The method's positioning is confusing—it is unclear whether FSSM targets the SSM community or signal processing community, and Section 3's presentation lacks intuition for either audience. The two application domains (radar, speech) use different architectures, raising questions about how general the approach truly is. Speech Commands V2 was criticized as too simple a benchmark; while AudioSet results were added in rebuttal, broader applicability remains uncertain. One reviewer (fi6o) explicitly stated they could not assess the paper due to specialized writing. The method appears to be a learnable filterbank implemented via SSM recurrences, but the novelty over existing learnable front-ends (SincNet, LEAF) is not clearly articulated.

**Reviewer Concerns:**

Addressed: (1) RFMamba comparison added showing substantial gains (F1: 0.83 vs 0.98); (2) AudioSet experiments with AST backbone added (mAP: 0.340 → 0.365); (3) Frozen vs learned bands ablation demonstrates frequency selectivity matters; (4) Mean ± std reported for RADIal; (5) Design-choice ablations added (phase init, learnable damping, stability); (6) Figures improved and proofreading done; (7) Signal processing background expanded. Outstanding: (1) No reviewer responded post-rebuttal due to reported OpenReview bug, so it's unclear if concerns were truly resolved; (2) Positioning remains confusing—fi6o could not assess the paper; (3) Generalizability beyond radar/speech is speculative; (4) The core novelty over SincNet/LEAF (which also learn filterbank parameters) remains unclear despite added comparisons; (5) Section 3 clarity issues were acknowledged but fundamental accessibility concerns persist.

**Reviewer Scores:**

fi6o (2, confidence 1): Explicitly stated unable to assess paper and requested AC seek other reviewers; added background may help but fundamental accessibility issue remains, score potentially 4 if convinced. mX8w (4): RFMamba comparison directly addressed their main concern; score potentially 4 or 6. QdMU (2): AudioSet experiments, frozen-band ablation, and statistical reporting addressed concerns; score potentially 4. cgvh (4): Positioning clarification, design ablations, and parallelization discussion addressed concerns; score potentially 4 or 6.

---

### Decision · Program_Chairs · 2026-01-26

Reject